# HotBEV: Hardware-oriented Transformer-based Multi-View 3D Detector for BEV Perception

Peiyan Dong[1]*, Zhenglun Kong[1]*, Xin Meng[1]*, Pinrui Yu[1], Yifan Gong[1], Geng Yuan[2],
Hao Tang[3]†, Yanzhi Wang[1]
[1]Northeastern University, [2]University of Georgia, [3]Carnegie Mellon University
{dong.pe, kong.zhe, yanz.wang}@northeastern.edu

## Abstract

The bird's-eye-view (BEV) perception plays a critical role in autonomous driving systems, involving the accurate and efficient detection and tracking of objects from a top-down perspective. To achieve real-time decision-making in self-driving scenarios, low-latency computation is essential. While recent approaches to BEV detection have focused on improving detection precision using Lift-Splat-Shoot (LSS)-based or transformer-based schemas, the substantial computational and memory burden of these approaches increases the risk of system crashes when multiple on-vehicle tasks run simultaneously. Unfortunately, there is a dearth of literature on efficient BEV detector paradigms, let alone achieving realistic speedups. Unlike existing works that focus on reducing computation costs, this paper focuses on developing an efficient model design that prioritizes actual on-device latency. To achieve this goal, we propose a latency-aware design methodology that considers key hardware properties, such as memory access cost and degree of parallelism. Given the prevalence of GPUs as the main computation platform for autonomous driving systems, we develop a theoretical latency prediction model and introduce efficient building operators. By leveraging these operators and following an effective local-to-global visual modeling process, we propose a hardware-oriented backbone that is also optimized for strong feature capturing and fusing. Using these insights, we present a new hardware-oriented framework for efficient yet accurate camera-view BEV detectors. Experiments show that HotBEV achieves a 2%~23% NDS gain, and 2%~7.8% mAP gain with a 1.1×~3.4× speedups compared to existing works on V100; On multiple GPU devices such as GPU GTX 2080 and the low-end GTX 1080, HotBEV achieves 1.1×~6.3× faster than others. The code is available at HotBEV.

## 1 Introduction

Recently, there has been a growing interest in 3D object detection (also known as BEV perception tasks) from multi-camera images, especially in the context of autonomous driving. While LiDAR-based methods have made remarkable progress, camera-only-based approaches have gained extensive attention due to their lower expenses and ability to identify color-based road elements such as traffic lights and signs [1]. Mainstream camera-only-based approaches for multi-camera 3D object detection tasks can be broadly classified into two research domains: 1) 3D reconstruction detectors, which project 2D features from the image view into the bird's-eye view (BEV) to extract intrinsic and extrinsic information [2, 3, 4, 5, 6, 7]. However, these approaches suffer from degradation due to the inaccuracy of the depth information predicted from 2D features. 2) 3D projection detectors,

---

*Equal Contribution
†Corresponding Author

37th Conference on Neural Information Processing Systems (NeurIPS 2023).

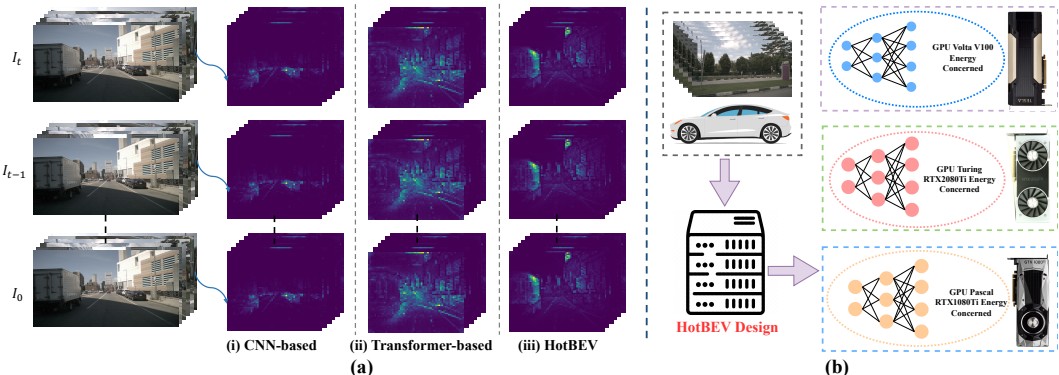

(i) CNN-based    (ii) Transformer-based    (iii) HotBEV

(a)                (b)

Figure 1: (a) Comparison with previous backbone methods in real-world object detection. Hot-BEV captures the global/local information better than others. (b) HotBEV aims to search for specialized model designs for target GPU devices.

which encode information such as 3D position and object queries into 2D features and then sample corresponding features for end-to-end 3D bounding box prediction [8, 9, 1]. These approaches overcome the problem of inaccuracies in the BEV features caused by the prediction of depth values or distributions. In particular, 3D projection detectors are considered one of the dominant future multi-view BEV perception techniques.

Despite the impressive performance exhibited by 3D projection detectors, their on-device inference speed is limited by various factors. i) Detectors employing a CNN-based backbone often demonstrate inferior detection performance compared to those with a transformer-based backbone [10]. Additionally, transformers are known to possess computation and memory complexities [11], which obstruct scalability, particularly when deploying them on resource-limited devices. ii) Many prior works [2, 7, 8] prioritized efficient model design by only relying on hardware-agnostic metrics like computation FLOPs, neglecting to account for the actual hardware's performance, such as the actual inference latency. Current detectors lack optimizations for specific hardware deployments, including factors like memory access cost, degree of parallelism, and compiler characteristics,

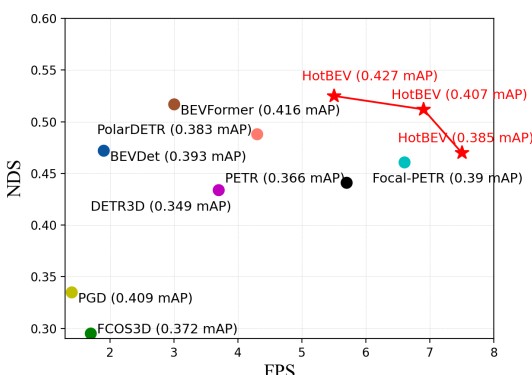

Figure 2: The trade-off between performance (mAP/NDS) and hardware efficiency (FPS) for different methods on the nuScenes test dataset. HotBEV outperforms SOTA efficient transformer-based detectors on both precision and efficiency. Models are evaluated on NVIDIA V100.

all of which significantly impact hardware throughput during inference. The quadratic memory complexity associated with original self-attention operations further exacerbates the disparity between theoretical FLOPs and actual hardware performance. Notably, most current in-vehicle chips adopt the GPU architecture, which poses greater challenges and requires further investigation despite its essential role. iii) To bridge the gap between the theoretical and practical efficiency of deep models, researchers have started considering the real latency in the network design phase. Hardware profiling [12, 13], which is generally utilized to estimate on-device speed, is dedicated and time-consuming. And profiling should be redone once the model structure, size, and hardware type change. Thousands of measurements are required for each operation to ensure data correctness. In contrast, there is a need for a practical, plug-and-play model capable of efficiently estimating inference latency, specifically designed for general GPU architectures.

In this paper, we propose a hardware-oriented transformer-based framework (HotBEV) for camera-only 3D detection tasks, which achieves both higher detection precision and remarkable speedup across both high-end GPUs and low-end GPUs (see Figure 1). Firstly, we propose a theoretical latency prediction model by considering the algorithm, the scheduling strategy, and the hardware properties. Given a target GPU, we directly use the latency, rather than the computation FLOPs, to guide our algorithm design. Then we perform a latency breakdown of major modules in classic camera-only detectors and figure out that the backbone is usually the speed bottleneck. After benchmarking

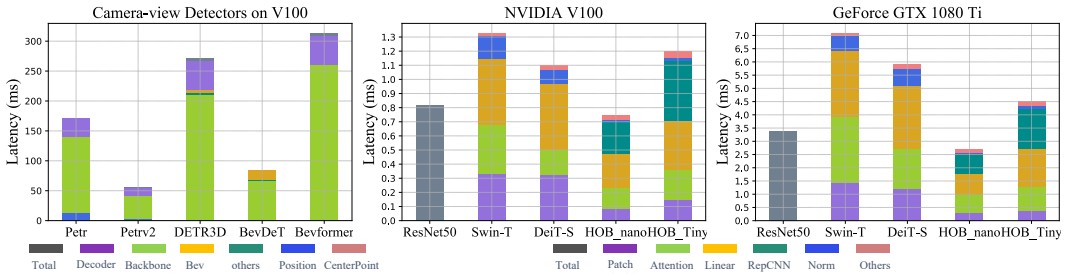

Figure 3: **Device speed breakdown**. Left: The latency breakdown of representative Camera-only Detectors (Input resolution: 3x256x704 for PETRv2, others 3x900x1600). Middle and Left: Results are obtained on NVIDIA V100, GeForce GTX 2080 Ti, and GeForce GTX 1080 Ti. The on-device speed for frequently used backbone and various operators is reported. (Input resolution: 3x224x224)

standard backbones on high-end (V100) and low-end (GTX 1080 Ti) GPUs, we propose efficient operators and fusion techniques for model on-device implementation. Based on these operators and the process of vision modeling, we design a hardware-oriented backbone with strong feature enhancement by information interaction between model stages. Then we extend the latency-aware design methodology to other parts, such as image embedding and decoder, and propose the basic design paradigm of HotBEV. Finally, guided by the latency prediction model, we generate the family of HotBEV through a standard search algorithm. Experiments show that our proposed framework can achieve a 2%~23% NDS gain and 2%~7.8% mAP with a 1.1×~3.4× speedups compared to existing works on V100 (see Figure 2). On multiple GPU devices such as GPU GTX 2080 HotBEV can reach 1.1×~6.3× faster than other models; for the low-end GTX 1080, our framework can achieve 1.1×~4.9× faster than others. Overall, our contributions are summarized as follows:

- We propose a latency-aware design methodology for 3D object detection based on BEV perception.
- We propose a hardware-oriented backbone that excels at capturing and fusing features.
- According to our analysis of efficient operators and strong feature modeling, we propose a new hardware-oriented framework for the BEV detector.
- We conduct experiments to showcase the superior inference accuracy of HotBEV compared to SOTA efficient BEV detectors while also highlighting its significant hardware efficiency.

## 2 Related Work

Current multi-view 3D object detectors can be divided into two schemas: LSS-based schema [3] and transformer-based schema.

**LSS-based schema**. BEVDet [2] is the first study that combines LSS and LiDAR detection head which uses LSS to extract BEV feature and LiDAR detection head to propose 3D bounding boxes. By introducing previous frames, BEVDet4D [5] acquires the ability of velocity prediction. For the above works, the models are complex, so many hardware platforms do not support some inside operators.

**Transformer-based schema**. BEVDepth [2] uses LiDAR to generate depth GT for supervision and encodes camera intrinsic and extrinsic parameters to enhance the model's ability of depth detection. DETR3D [9] extends DETR [14] into 3D space, using a transformer to generate 3D bounding boxes. Following the DETR3D, PETR [8] converts 3D coordinates into 3D position embedding to perceive the 3D scene. And BEVFormer [1] uses deformable transformer [15] to extract features from images and cross attention to link the characteristics between different frames for velocity prediction. Although better performance is achieved through these works, the computation cost (e.g., over 170 GFLOPs) or realistic speed has yet to be optimized. We are the first to work on transformer-based 3D detectors to explore comparable performance and fast inference speed on diverse hardware platforms while maintaining acceptable detection accuracy.

**Hardware-aware network design.** Several existing works have addressed the issue of realistic latency during the network design phase, exploring two distinct directions. Firstly, some researchers evaluate speed directly on targeted devices and derive guidelines for developing hand-designed efficient models, as demonstrated by [16]. Secondly, others employ the technique of neural architecture search (NAS) to search for fast models, exemplified by [17]. Nevertheless, conducting speed tests on various hardware platforms for our proposed structures can be exceedingly labor-intensive due to the extensive range of candidates and their corresponding properties. Additionally, obtaining precise

results for small-scale structures on low-end devices presents its own set of challenges. In contrast, our latency prediction model directly estimates the inference speed on specific computing platforms, circumventing these difficulties.

## 3    Methodology

In this section, we present our approach to efficiently and accurately predict the on-device latency of an architecture on the target device using the latency prediction model. Additionally, we detail efficient operators and fusion implementation techniques. According to the process of vision modeling, we introduce an efficient embedding module and a hardware-oriented backbone that excels at capturing and fusing features. Then our proposed basic design paradigm for HotBEV is based on these efficient operators, the process of vision modeling, and the detection flow of DETR3D [9]. Finally, guided by the latency prediction model, we generate a family of models known as HotBEV through a standard search algorithm [18]. Note that we also adapt the Temporal Aligned Module (TAM) in [19] to efficiently improve performance and robustness.

### 3.1    Latency-aware Design

#### 3.1.1    Latency Prediction Model

We introduce a novel latency prediction model, denoted by $E$, which enables direct prediction of the latency of runtime design choices on any target GPU. This allows for the efficient identification of optimal model settings on corresponding platforms. Specifically, the latency predictor $E$ considers the hardware properties $H$, layer type $T$, channel dimension $C$, and input granularity $G$ as input for each design choice and outputs the predicted latency of the block as $l = E(H, T, C, G)$.

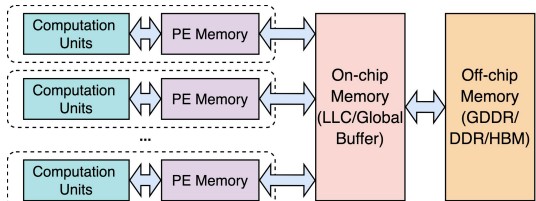

Figure 4: **Hardware Modeling**. The memory system includes 1) off-chip memory, 2) on-chip global memory, and 3) memory in the PEs.

**Hardware model.** We model a hardware device as multiple processing engines (PEs), allowing for parallel computation with varying degrees of parallelism. As illustrated in Figure 4, we represent the memory system using a three-level structure [20, 21], which includes: 1) off-chip memory, 2) on-chip global memory, and 3) memory in the PEs. This hardware model enables accurate prediction of the latency of data movement and computation.

**Latency prediction modeling.** It includes three steps: 1) *Input/output shape definition.* The initial step is to calculate the input and output shapes. 2) *Operation mapping to hardware.* Based on our hardware model, we first divide the output feature map into multiple tiles and assign each tile to a separate PE for parallel processing. 3) *Latency estimation.* We evaluate the latency of each tile, which comprises both data movement and computation latency: $l = l_{data} + l_{compute}$. For $l_{data}$, we leverage our hardware model (see Figure 4) and compute the sum of input and output data movement latencies as $l_{data} = l_{in} + l_{out}$. These latencies are estimated based on hardware bandwidth and input granularity $G$ (equivalent to resolution scale). We assume that each PE moves the required input feature maps and weights just once to compute an output tile. For $l_{compute}$, we use the maximum throughput of FP32 computation on PEs and the FLOPs required to compute a single output tile. The total computation latency can be determined by considering the number of tiles and PEs. We test three types of hardware devices, NVIDIA V100, NVIDIA GTX 2080, and NVIDIA GTX 1080. For a more detailed description of our latency prediction model, please refer to Appendix A.

#### 3.1.2    Efficient Operators and Fusion Techniques of Model Implementation

The development of efficient network architectures for resource-limited devices has greatly benefited from reduced parameters and floating-point operations (FLOPs) and improved accuracy. However, conventional efficiency metrics, such as FLOPs, overlook memory cost and degree of parallelism. In this study, we aim to improve network runtime speed and detection performance by identifying and modifying building blocks that are not hardware-friendly. To achieve this, we first perform a latency breakdown of major modules in classic camera-only detectors, DETR3D, PETR [8], PETRv2 [19],

BEVDeT [4], and BEVFormer [1], which reveals that the backbone is usually the speed bottleneck (66%∼78%). Therefore, we deploy common backbones on high-end (V100) and low-end (GTX 1080 Ti) GPUs and benchmark their speed, as illustrated in Figure 3. We then introduce efficient operators and fusion techniques to improve the speed and detection performance of these networks.

**Convolutional modulation for efficient global modeling.** Instead of calculating the similarity score matrix (attention matrix [22]), we simplify self-attention by modulating the value $V$ with convolutional features as Figure 5(c). Our approach uses convolutional modulation rather than self-attention to build relationships since they are more memory-efficient, particularly when processing high-resolution images. Due to the modulation operation, our method differs from traditional residual blocks and can adapt to the input content.

**Fusing BatchNorm into the preceding fully-connected layer.** After analyzing various backbones as shown in Figure 3, we found that LayerNorm (LN) accounts for approximately 10% to 15% of the network's total latency. Dynamic normalization techniques like LN gather running statistics during inference, resulting in delays that impact speed. On the other hand, BatchNorm (BN) is more memory and computation-efficient, as it is fused with the preceding fully-connected layer, reducing data movement and computation load (see Figure 4). To this end, we modified the WMSA/SWMSA in Swin [10] by replacing LN-Linear with Linear-BN, which we refer to as WMSA$_{bn}$/SWMSA$_{bn}$. By switching to a BN-based design, we achieved a 13% to 15% speedup with no significant loss in accuracy (less than 0.3%). Based on the accuracy analysis in Appendix A, one advantage of the window-based self-attention on the efficient implementation compared to other transformer architectures: It can replace the LayerNorm with BatchNorm and then perform the fusing technique with acceptable accuracy. And replacing the LayerNorm directly with the BatchNorm in the original self-attention is difficult since the model training cannot converge. Therefore, we consider WMSA$_{bn}$/SWMSA$_{bn}$ as our design candidates.

**Fusing multiple branches into one single branch in reparameterized CNNs.** Multi-branch structures come with increased data movement cost, as the activation values of each branch are saved into PE memory or on-chip memory (if the PE memory is insufficient) to compute the subsequent tensor in the graph. Additionally, the synchronization cost arising from multiple branches impacts the overall runtime [23]. To address these challenges, we use RepCNN [24] as a network component, which fuses multiple branches into more single-branch substructures during inference. This approach enables even distribution of computation among multiple PEs, preventing imbalanced computation overheads associated with multiple branches. The resulting operator fusion improves memory access and parallel computation on multiple PEs.

## 3.2 Architecture Design

We present HotBEV, a hardware-oriented framework for multi-view 3D detection. Given a set of $I=\{I^i \in \mathbb{R}^{3 \times H \times W}, i = 1, \ldots, N\}$, our framework leverages a hardware-oriented backbone (HOB) to extract 2D multi-view features, $F^{2d}=\{F_i^{2d} \in \mathbb{R}^{C \times H \times W}, i = 1, \ldots, N\}$. The *coordinates generator* generates 3D coordinates, which are aligned with the coordinate system of the frame $t$ using the temporal aligned module (TAM) concerning the previous frame $t - 1$. Next, the 2D features and 3D coordinates from adjacent frames are concatenated and fed into the 3D *position encoder* to obtain 3D-aware features. Our hardware-oriented decoder uses these features as keys and values, which interacts with detection queries initialized from standard learnable 3D anchor points with a small MLP network. Finally, the updated queries are input to the detection head for the final prediction. For *coordinates generator* and *position encoder*, please refer to [8].

### 3.2.1 Hardware-Oriented Backbone with Strong Feature Enhancement

**Hardware-oriented backbone.** Based on our GPU breakdown 3, we found that the backbone consistently caused the most latency. To maintain detection precision while removing this speed bottleneck, we propose a powerful transformer encoder for feature capturing and fusing, as shown in the backbone design in Figure 5. We divide the backbone into four stages $S$, following the granularity of data flow in ResNet [25]. Since features scale from the local to the global visual receptive field, we introduce the HOB block design as shown in Figure 5(a). In each HOB block, we use several consecutive local-wise attention mechanisms to extract local information (texture-level semantics), followed by global attention to enhance global information (abstract-level semantics) in the feature map. Furthermore, we insert a semantic-augmented module (consisting of an upsampling layer

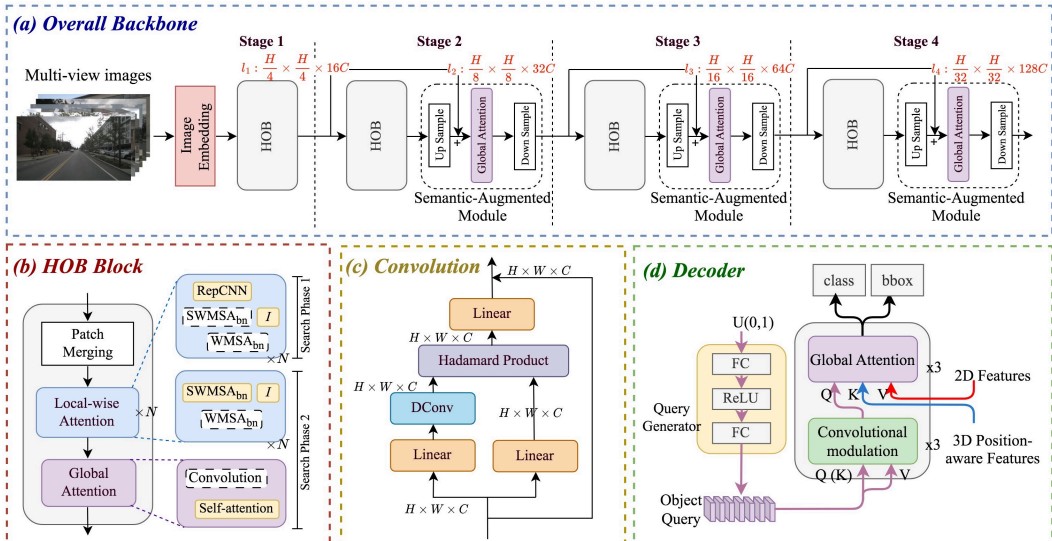

Figure 5: **The design of our proposed HotBEV.** The workflow follows the DETR3D, and the main components include that (a) Overall backbone: includes four stages, where feature size scales along the stage. To enhance the semantic information of low-level features, one semantic-augmented module (SAM) is added at the end of each stage. (b) Design of the HOB block, which contains multiple local/global attention layers. (c) The convolutional modulation for efficient global modeling. (d) Hardware-Oriented Decoder.

and global attention) into every two consecutive HOB blocks (except between Stage 1 and 2) to further enhance low-level semantics in the current stage, as shown in Figure 12. Notably, unlike other methods such as [10, 25], we simultaneously enhance texture-level and abstract-level information in each stage. To reinforce texture-level semantics, we leverage information interaction between stages, not just between layers in the same stage. By using the efficient operators described in Section 3.1.2, we provide the "two-phase design space" (DS) of the HOB backbone:

$$
\begin{aligned}
\text{DS}^1_{i,local,s=1,2,3,} &\in \{\text{RepCNN}^i, \text{WMSA}^i_{bn}, \text{SWMSA}^i_{bn}\}, \\
\text{DS}^2_{i,local,s=4,} &\in \{\text{WMSA}^i_{bn}, \text{SWMSA}^i_{bn}\}, \\
\text{DS}^{1,2}_{i,glocal} &\in \{\text{ConvModula}\},
\end{aligned}
\tag{1}
$$

where *local* represents the candidates of local-wise attention while *global* for the global attention; the 1st phase covers $S_1$, $S_2$, $S_3$ of the backbone, and the 2nd phase for $S_4$; $i$ denotes the $i^{th}$ block.

**Image embedding module.** Figure 3 reveals that patch embedding is a speed bottleneck on multiple platforms due to its reliance on a non-overlapping convolutional layer with large kernel size and stride. Unfortunately, most compilers and acceleration techniques, such as Winograd, do not support this type of layer well. To address this issue, we propose a faster downsampling method using a convolution stem, which consists of three hardware-efficient $3\times3$ convolutions with a stride of 2. To obtain input embeddings $L_0$ of size $\frac{H}{G} \times \frac{W}{G} \times C$ for an input image $x \in \mathbb{R}^{H \times W \times 3}$, we divide it into $\frac{H \times W}{G \times G}$ patches and feed them to the convolution stem:

$$
l_0^{\frac{H}{G} \times \frac{W}{G} \times D_1} = \text{PatchEmbed}(x^{H \times W \times 3}).
\tag{2}
$$

### 3.2.2 Hardware-Oriented Decoder

To alleviate the convergence difficulties in the 3D scene, we initialize a set of learnable 3D anchor points with a 0~1 uniform distribution. Then, we input the coordinates of the 3D anchor points into a small MLP network with two linear layers of Sigmoid function in between and generate the initial object query $Q_0$. The decoder is applied to predict the final abstract semantics and generate the bounding box. Our decoder consists of six global attention blocks and is divided into two parts, as shown in Figure 5(d). The front three layers take object queries as the only input and perform the self-attention computation, aiming at separating different objects as [14]; The back three layers take object queries as queries and image features as keys and values. As shown in Figure 12, they perform the cross-attention between object queries and image features to extract the content and position of

Table 1: Comparison of different methods on the nuScenes val set. FPS is tested on V100 with INT8.

| Method | Backbone | Resolution | Frames | NDS ↑ | mAP ↑ | mATE ↓ | mASE ↓ | mAOE ↓ | mAVE ↓ | mAAE ↓ | FPS |
|---|---|---|---|---|---|---|---|---|---|---|---|
| BEVDet | ResNet50 | 256 × 704 | 1 | 0.379 | 0.298 | 0.725 | 0.279 | 0.589 | 0.86 | 0.245 | 25.8 |
| BEVDet4D | ResNet50 | 256 × 704 | 2 | 0.457 | 0.322 | 0.703 | 0.278 | 0.495 | 0.354 | 0.206 | 25.9 |
| PETRv2 | ResNet50 | 256 × 704 | 2 | 0.456 | 0.349 | 0.7 | 0.275 | 0.58 | 0.437 | 0.187 | 30.2 |
| BEVDepth | ResNet50 | 256 × 704 | 2 | 0.475 | 0.351 | 0.639 | 0.267 | 0.479 | 0.428 | 0.198 | 24.3 |
| FastBEV-MS | ResNet50 | 256 × 704 | 4 | 0.485 | 0.343 | 0.647 | 0.282 | 0.36 | 0.342 | 0.225 | – |
| SOLOFusion | ResNet50 | 256 × 704 | 16 | 0.534 | 0.427 | 0.567 | 0.274 | 0.511 | 0.252 | 0.181 | 20.2 |
| SOLOFusion | ResNet50 | 256 × 704 | 4 | 0.494 | 0.362 | 0.607 | 0.304 | 0.539 | 0.293 | 0.19 | 21.4 |
| **HotBEV** | HOB-nano | 256 × 704 | 4 | 0.455 | 0.35 | 0.636 | 0.271 | 0.449 | 0.542 | 0.36 | **39.2** |
| **HotBEV** | HOB-tiny | 256 × 704 | 4 | 0.487 | 0.362 | 0.628 | 0.27 | 0.438 | 0.412 | 0.19 | 31.7 |
| **HotBEV** | HOB-base | 256 × 704 | 4 | 0.506 | 0.369 | 0.625 | 0.264 | 0.362 | 0.364 | 0.153 | 27.2 |
| BEVDet | ResNet101-DCN | 640 x 1600 | 1 | 0.472 | 0.393 | 0.608 | 0.259 | 0.366 | 0.822 | 0.191 | 2.95 |
| FCOS3D | ResNet101-DCN | 900 × 1600 | 1 | 0.395 | 0.372 | 0.806 | 0.268 | 0.511 | 1315 | 0.7 | 2.64 |
| DETR3D | ResNet101-DCN | 900 × 1600 | 1 | 0.434 | 0.349 | 0.716 | 0.268 | 0.379 | 0.842 | 0.2 | 5.74 |
| PGD | ResNet101-DCN | 900 × 1600 | 1 | 0.335 | 0.409 | 0.732 | 0.263 | 0.423 | 1.285 | 0.172 | 2.17 |
| Focal-PETR | ResNet101-DCN | 512 × 1408 | 1 | 0.461 | 0.39 | 0.678 | 0.263 | 0.395 | 0.804 | 0.202 | 10.23 |
| PETR | ResNet101-DCN | 512 × 1408 | 1 | 0.441 | 0.366 | 0.717 | 0.267 | 0.412 | 0.834 | 0.19 | 8.84 |
| BEVFormer | ResNet101-DCN | 900 × 1600 | 4 | 0.517 | 0.416 | 0.673 | 0.274 | 0.372 | 0.394 | 0.198 | 4.65 |
| PolarDETR | ResNet101-DCN | 900 × 1600 | 2 | 0.488 | 0.383 | 0.707 | 0.269 | 0.344 | 0.518 | 0.196 | 5.43 |
| **HotBEV** | HOB-nano | 512 × 1408 | 4 | 0.47 | 0.385 | 0.648 | 0.243 | 0.422 | 0.715 | 0.183 | 12.38 |
| **HotBEV** | HOB-tiny | 512 × 1408 | 4 | 0.512 | 0.407 | 0.634 | 0.235 | 0.408 | 0.632 | 0.175 | 11.39 |
| **HotBEV** | HOB-base | 512 × 1408 | 4 | **0.525** | **0.427** | **0.62** | **0.221** | **0.36** | **0.55** | **0.163** | 9.08 |

the object. For efficient modeling, we leverage the convolutional modulation layer as the decoder component (more details in Appendix A).

To generate temporal-aligned 3D coordinates, we adapted the technique TAM [19]. Subsequently, we combine the resulting 3D coordinates with the 2D features and feed them into a 3D *position encoder* [8]. This allows us to obtain temporal-aligned 3D-aware features, which improve the model's localization, attitude, speed estimation, and overall robustness.

## 3.3   Training

Our design space (Eq. (1)), also as search space is a selection of possible blocks, including RepCNN, WMSA, SWMA (for local-wise attention layer), and convolutional modulation (for global-wise attention layer). We propose a simple, fast yet effective gradient-based search algorithm to obtain a candidate network that just needs to train the supernet for once. To train our supernet, we adopt the Gumble Softmax sampling to get the importance score for the blocks within each search space/stage. During each step of training, a number of blocks are sampled to obtain a subnet structure. The latency of this subnet can be estimated using our latency prediction model.

**Supernet design.** We use the *two-phase design space* (DS) as the search space and train the supernet for the HOB backbone. We only search the backbone's structures, dimensions $C$, and input granularity $G$, while the decoder uses fixed structures with dimensions adapted to the backbone.

**Latency-aware model slimming.** It has three steps:

1) Train the supernet with the Gumble Softmax sampling [26] to get the importance score for the blocks within each DS.

2) Use latency prediction model $E$ (Section 3.1.1) to estimate the on-device latency of each candidate.

3) Perform latency-aware model slimming on the supernet obtained from step 1) by FPS evaluated with predictor $E$. Specifically, we use the score $s$ of each candidates to define the importance score of $DS_i$ as $\frac{s_i^{\text{RepCNN}} + s_i^{\text{WMSA}}}{s_i^{\text{SWMSA}}}$ for $S_1$, $S_2$, $S_3$, and $\frac{s_i^{\text{WMSA}}}{s_i^{\text{SWMSA}}}$ for $S_4$. We sum up all the scores of all DS within that $S$ and deduce the score for each $S$. Then we define the evolution process (all performed in the current least important $S$): a) remove the $1^{st}$ SWMSA ; b) remove the $1^{st}$ WMSA; c) reduce the width by multiples of 16. Then we predict the current FPS $f$ and decide by FPS_NDS_drop(-%*f). This process is repeated until the target throughput is reached.

## 4   Experiments

### 4.1   Datasets and Implementation Details

We conduct comprehensive experiments on the nuScenes dataset [27], which contains 1000 driving scenes of 20-second length for each. The scenes are officially split into 700, 150, and 150 scenes for training, validation, and testing. The dataset includes approximately 1.4M camera images.

Table 2: Comparison for large backbone and resolutions on the nuScenes val set (FPS on V100).

| Method | Backbone | Resolution | NDS ↑ | mAP ↑ | mATE ↓ | mASE ↓ | mAOE ↓ | mAVE ↓ | mAAE ↓ | FPS |
|---|---|---|---|---|---|---|---|---|---|---|
| PETR | ResNet101 | 1600×900 | 0.442 | 0.37 | 0.711 | 0.267 | 0.383 | 0.865 | 0.201 | 5.7 |
| PETRv2 | ResNet101 | 1600×640 | 0.524 | 0.421 | 0.681 | 0.267 | 0.357 | 0.377 | 0.186 | - |
| BEVDet4D | Swin-B | 1600×640 | 0.515 | 0.396 | 0.619 | 0.26 | 0.361 | 0.399 | 0.189 | - |
| BEVDepth | ResNet101 | 512×1408 | 0.535 | 0.412 | 0.565 | 0.266 | 0.358 | 0.331 | 0.19 | 2.3 |
| BEVFormerv2 | ResNet50 | 1600×640 | 0.529 | 0.423 | 0.618 | 0.273 | 0.413 | 0.333 | 0.181 | - |
| PolarFormer-T | ResNet101 | 1600×900 | 0.528 | 0.432 | 0.648 | 0.27 | 0.348 | 0.409 | 0.201 | 3.5 |
| Sparse4D | ResNet101-DCN | 900×1600 | 0.541 | 0.436 | 0.633 | 0.279 | 0.363 | 0.317 | 0.177 | 4.3 |
| **HotBEV** | HOB-base | 512 × 1408 | **0.525** | **0.427** | **0.62** | **0.221** | **0.36** | **0.55** | **0.163** | 5.5 |

Evaluation metrics include mean Average Precision (mAP) and five types of true positive metrics (TP metrics): mean Average Translation Error (mATE), mean Average Scale Error (mASE), mean Average Orientation Error (mAOE), mean Average Velocity Error (mAVE), mean Average Attribute Error (mAAE). We also report nuScenes Detection Score (NDS) to capture all aspects of the nuScenes detection tasks. We follow the training recipe from PETR but mainly report results with 24 training epochs to compare with other detection models. Experiments were run on 8 NVIDIA V100.

## 4.2 Model Accuracy and Speed Performance

**Main results.** As delineated in Table 1, our model is compared to an array of existing camera-based methodologies. These include FCOS3D [28], DETR3D [9], PGD [29], PETR [8], PETRv2 [19], Focal-PETR [30], BEVDet [4], BEVDet4D [5], BEVFormer [1], BEVDepth [2], and PolarDETR [31]. The table lists the backbone type, image resolution, number of frames, inference speed (FPS), and accuracy on the nuScenes validation set for each method. The backbone options include ResNet50 and ResNet101-DCN [25]. Our streamlined models surpass other methods in both performance score and inference speed.

In particular, our compact model, HOB-tiny (256 × 704), with 0.487 NDS, 0.362 mAP, and 19.8 FPS, outperforms BEVDet (0.379 NDS, 0.298 mAP, 16.7 FPS), BEVDet4D (0.457 NDS, 0.322 mAP, 16.7 FPS), PETRv2 (0.456 NDS, 0.349 mAP, 18.9 FPS), and BEVDepth (0.475 NDS, 0.351 mAP, 15.7 FPS) with a ResNet50 backbone, achieving a 2.5% ∼ 28.5% NDS gain, 3.1% ∼ 21.5% mAP gain and 4.8% ∼ 26.1% FPS gain. The reductions in the orientation error, velocity error, and attribute error all contribute to enhancing the NDS score. Even with a larger input resolution, our model continues to offer an unrivaled balance of accuracy and speed compared to extant work. For instance, our HOB-nano model (512 × 1408) achieves 2% NDS and 13.6% FPS increase with a 1.3% mAP and score difference when compared to Focal-PETR (ResNet101-DCN, 512 × 1408). Our HOB-tiny model (512 × 1408) surpasses PolarDETR (0.488 NDS, 0.383 mAP, 3.5 FPS) in both speed and accuracy. Our HOB-base model (512 × 1408) also surpasses BEVFormer (0.517 NDS, 0.416 mAP, 3 FPS) in both speed and accuracy. For detailed configurations of our model and detailed analysis of temporal modeling and robustness related to frame length, please refer to Appendix A.

Our research specifically targets small models, so our results are particularly favorable for these models compared to other studies. For a comprehensive understanding, we also present a comparison with baseline models that possess larger backbones and increased input sizes in Table 2. Notably, we surpass baseline models in frames per second (FPS) while maintaining comparable accuracy levels.

**Comparison with 3D reconstruction-based BEV detectors.** Table 3 shows that 3D reconstruction-based BEV detectors are better than our methods by up to 19.8 NDS but with inferior on-device speed than ours by up to 19.4 FPS. Even though the detection performance of image-only detectors cannot be superior to Lidar-only methods, these results also motivate us to implement our design into one of the commercial mainstream, post-fusion systems to improve hardware efficiency. Currently, the pre-fusion family, such as BevFusion, is costly to deploy in practice. The current in-vehicle market is dominated

Table 3: Comparison with 3D reconstruction-based BEV detectors on the nuScenes val set.

| Methods | NDS↑ | mAP↑ | FPS |
|---|---|---|---|
| PointPillars | 61.3 | 52.3 | 29 |
| SECOND | 63 | 52.6 | 14.3 |
| CenterPoint | 66.8 | 59.6 | 12.4 |
| HotBEV-nano | 47 | 38.5 | 31.8 |
| HotBEV-tiny | 51.2 | 40.7 | 20.4 |
| HOB-base | 52.5 | 42.7 | 16.1 |

by camera-only solutions, such as Tesla, or post-fusion of camera and lidar signals, such as XPeng Motors, whose camera part still deploys camera-only solutions [32]. Hence, our research aims to explore the efficient and effective camera-only 3D detection series for practical implementation. So we choose state-of-the-art frameworks, and multi-frame models as the starting point of our research.

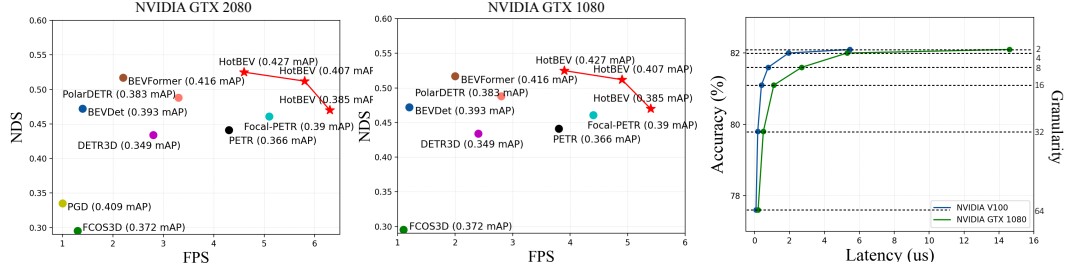

Figure 6: **Left and middle**: The trade-off between performance (NDS) and hardware efficiency (FPS) for different detection methods on the nuScenes val set with different GPUs. **Right**: Various granularity settings on the HOB backbone of HotBEV.

Moreover, according to [33], the camera processing modules occupy $33\% \sim 42\%$ of the whole latency distribution for the post-fusion system. Our hardware-oriented design with enhanced image feature representation can be leveraged in the camera encoder part of the current in-vehicle detection system to improve on-device efficiency.

**Performance on multiple GPU devices.** To evaluate the hardware throughput, we implement the latency-aware model slimming on two other devices: NVIDIA GTX 2080 Ti which has a 5.5M size L2 cache with 448 Gbps memory bandwidth; NVIDIA GTX 1080 Ti which has a 5.5M size L2 cache with 325 Gbps memory bandwidth; We report the average FPS of over 1000 inferences. As depicted in Figure 6, our approach surpasses current camera-only BEV frameworks in terms of both hardware efficiency and performance. Other methods typically overlook the constraints imposed by limited memory and parallelism in on-device runtimes, leading to further degradation in speed performance on devices with limited resources. Without hardware optimization, e.g., int8 quantization, some methods fail to produce results within a reasonable timeframe. In contrast, our models strike the optimal balance between speed and performance, making them the superior choice among existing approaches. For example, on GTX 2080, our HotBEV (0.385 mAP) is $4.5\times$ faster than BEVDet (0.393 mAP); our HotBEV (0.407 mAP) is $2.6\times$ faster than BEVformer (0.416 mAP); our HotBEV (0.427 mAP) is $1.4\times$ faster than PolarDETR (0.383 mAP).

The speed-up effect is superior to that in Table 1, demonstrating our framework's enhanced GPU generalization ability compared to other approaches. It's worth noting that our logic extends beyond autonomous driving. Firstly, our backbone is designed to cater to general vision tasks. Secondly, the hardware model is specifically optimized for GeMM. In the appendix, we showcase our implementation on diverse general vision tasks such as classification and 2D detection.

**Performance on multiple commercial Orin.** It is necessary to test the speed on an actual commercial chip. As shown in Table 4, we test our HotBEV models on Orin to validate our framework. Before testing, we quantized our model into INT8 with a tensorRT engine. And then run the test 50 times for each model to obtain stable results.

Table 4: Our proposed results on Orin.

| Methods | Backbone | Resolution | NDS↑ | mAP↑ | FPS |
|---------|----------|------------|------|------|-----|
| HotBEV | HOB-nano | 512x1408 | 0.47 | 0.385 | 31.8 |
| HotBEV | HOB-tiny | 512x1408 | 0.512 | 0.407 | 20.4 |
| HotBEV | HOB-base | 512x1408 | 0.525 | 0.427 | 16.1 |

### 4.3 Ablation Study

In this section, we conduct the ablations with HotBEV-nano trained 24 epochs. The backbone is pre-trained on ImageNet dataset [34] and trained on Nuscenes. The input image size is $256\times704$, and the number of detection queries is set to 900.

Table 5: Ablation study of HOB.

| Index | GA | SAM | NDS | mAP | FPS |
|-------|----|----|------|------|-----|
| ① | - | - | 0.396 | 0.328 | 21.3 |
| ② | ✓ | - | 0.443 | 0.337 | 21.4 |
| ③ | ✓ | GA | 0.447 | 0.340 | 23.7 |
| ④ | ✓ | ✓ | **0.455** | **0.350** | **24.5** |

#### 4.3.1 Analysis of Components in HotBEV

**Major components of HOB.** Table 5 studies how global attention (GA) and semantic-augmented module (SAM) contribute to HOB performance. We only modify the backbone network without disabling the 3D position encoder, TAM, and decoder modules. ① and ② show that inserting our GA after the local-wise attention can improve 0.9% mAP and 4.7% NDS without significant impact on speed. ② and ③ show that inserting one GA block costs 2.3 FPS yet only gains 0.3% mAP. Once we replace the GA with the SAM, mAP is increased from 34% to 35%, while NDS is increased from 44.7% to 45.5%. So SAM can enhance performance and be better than simple global modeling.

### 4.3.2 Analysis of Prediction Modeling

**Empirical validation.** To evaluate the prediction model, we varied the input granularity $G$ and used the 1-*st* block in Swin-T as a case study. Figure 7 compares our predictions and the actual testing latency on two different GPUs, the NVIDIA V100 and GTX 1080 Ti. The results demonstrate that the prediction model can accurately estimate the actual latency across a wide range of input granularities.

**More granularities settings.** We test various granularity settings on the HOB backbone of HotBEV-nano to examine the effects of $G$. The results on the Tesla-V100 GPU are presented in the left sub-figure of Figure 6. We increase $G$ from 2 and 64, which consistently improves the realistic efficiency of both the NVIDIA V100 and GTX 1080 GPU. It can be found that the finest granularity ($G$=2) causes substantial inefficiency despite the mAP improvement. Otherwise, the coarsest granularity ($G$=64) benefits the speedup with large detection precision degradation. In specific, all networks are transformed into fixed matrix operations on GPU platforms (General Matrix Multiply, GeMM). The latency prediction model evaluates the speed of matrix multiplication and the corresponding data movement, so it is applicable to general networks. For example, this modeling can be generalized in other networks as the prediction results on the 1st block in ResNet-50 in Appendix A.

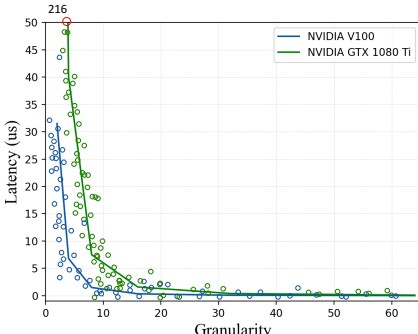

Figure 7: **Latency prediction results.** Results are tested on NVIDIA V100 and GTX 1080 Ti.

**More discussion on latency predictor.** Our proposed latency predictor provides some opportunities. (1) Efficient model generation, which also proceeds with AI democratization. As the benchmarking-based approach needs one-day training, our proposed theoretical latency predictor is training-free. For example, the benchmarking-based approach requires 5 days to generate the dataset of 5 different devices if 5 target models are demanded. In contrast, our proposed is off-the-shelf. Our method provides the opportunity for inexpensive and efficient research for users who do not have access to target devices. For instance, when the in-vehicle Orin chip is not accessible, the related efficient model research on the Orin chip can still be advanced. In conclusion, our approach makes sense for today's rapidly growing demand for autonomous driving. (2) The proposed latency predictor focuses on modeling the latency of Matrix Multiplication (MM) with generalizability. Indeed, strong GPU simulators cannot accurately model the behavior of the latest NVIDIA GPUs.' However, our purpose is not to describe the behavior of GPUs. We want to reflect on the relative performance of latency for different layer types and sizes on target GPUs. This is because our search goal is to minimize the relative time in the search space of the current device. For generalizability, our design focuses on latency modeling of MM, the typical computation operation in DNNs, which is mainly impacted by the computing performance of Tensor Core, not other specific operators, so the proposed predictor has generalizability, as shown in Figure 7 of our paper.

## 5 Conclusions and Limitations

We present a hardware-oriented transformer-based framework (HotBEV) for camera-only 3D detection tasks, which achieves higher detection precision and remarkable speedup across high-end and low-end GPUs. Firstly, we propose a theoretical, plug-and-play latency prediction model. Given a target GPU, we directly use the latency to guide our algorithm design. Based on the latency breakdown of classic camera-only detectors, we identify the backbone as the main speed bottleneck. Then, we propose efficient operators and fusion techniques for model on-device implementation. Based on these operators and the process of vision modeling, we design a hardware-oriented backbone with strong feature enhancement. Then we propose the basic design paradigm of HotBEV. Finally, guided by the latency prediction model, we generate the family of HotBEV through a standard search algorithm. Experiments show the superior inference accuracy of HotBEV compared to SOTA BEV detectors with significant on-device speed. Notably, while our primary focus lies in the camera-only method for BEV perception, our latency-aware design methodology can also be applied to fusion-based BEV methods, enabling efficient algorithm design that aligns with real-world requirements.

## 6 Acknowledgment

This work is supported in part by National Science Foundation CCF-1937500.

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
