# A  Appendix

## A.1  Latency prediction model

### A.1.1  Hardware Latency

In order to build our latency prediction model, We test three types of hardware devices, NVIDIA V100, NVIDIA GTX 2080, and NVIDIA GTX 1080. Their respective properties are presented in Table 6. It shows that the server GPU V100 is the most powerful hardware device with the most processing engines (#PE). Therefore, the computation with quadratic memory complexity, e.g., self-attention, could easily fall into a memory-bounded operation on V100 because of its high parallelism.

Table 6: **Hardware properties.**

| NAME | PE | FP32 | FREQUENCY(MHZ) | BANDWIDTH (G) |
|---|---|---|---|---|
| NVIDIA V100 | 80 | 64 | 1390 | 690 |
| NVIDIA GTX 2080 | 20 | 64 | 1710 | 325 |
| NVIDIA GTX 1080 | 20 | 64 | 1710 | 325 |

### A.1.2  Latency Prediction Modeling

Following Section 3.1.1, the inputs of the latency prediction model include: 1) the structure configuration of a candidate block, 2) the spatial granularity $G$, 3) the channel dimension $C$, and 4) the hardware properties are shown in Table 6. The latency of a candidate block is predicted according to the following three steps.

**Input/output shape definition.** Calculating the input and output shapes is the first step in determining an operation's latency. Taking the MSA operation as an example, the input of this operation is the activation with the shape of $C_{in} \times H \times W$, where $C_{in}$ is the number of input channels, and $H$ and $W$ are the resolutions of the feature map. The shape of the output tensor is $\frac{H}{G} \times \frac{W}{G} \times C_{out}$, where $\frac{H}{G} \times \frac{W}{G}$ is the number of output patches, $C_{out}$ is the number of output channels and $G$ is the spatial granularity.

**Operation-to-hardware mapping.** We map the operations to hardware. We have modeled a hardware device as multiple processing engines (PEs). We first consecutively split the output feature map into multiple tiles. Specifically, the shape of each tile is TP × TC × TS1 × TS2 ($\frac{H}{G} \times \frac{W}{G}$)×$T_C$×$T_G$×$T_G$. These split tiles are assigned to multiple PEs. The computation of each tile is executed in a PE.

**Latency estimation.** We evaluate each tile's latency, including the data movement latency and the computation latency: $l = l_{data} + l_{compute}$.

1) *Data movement latency $l_{data}$.* We model the memory system of hardware as a three-level architecture [20]: off-chip memory, on-chip global memory, and local memory in PE. The input data and weight data first move from the off-chip memory to the on-chip global memory. To simplify the latency prediction model, we assume that the hardware can fully utilize the off-chip memory bandwidth.

The data used to calculate the output tiles is moved from the on-chip global memory to each PE's local memory. The latency of data movement to local memory is estimated by its bandwidth and efficiency. To make the prediction model simpler, we assume that each PE only moves the appropriate input feature maps and weights once to compute an output tile. The time from off-chip memory to on-chip global memory and the time from on-chip global memory to local memory are added together to compute the input data movement latency $l_{in}$: $l_{in} = l_{off2on} + l_{global2local}$. The output data are transferred from local memory to on-chip global memory and subsequently to off-chip memory, in contrast to the input data: $l_{out} = l_{local2global} + l_{on2off}$. By combining the input and output data movement latency, we can determine the overall data movement latency: $l_{data} = l_{in} + l_{out}$.

The granularity $G$ impacts the latency of data movement because when it is small, more input data will be transferred to numerous PEs to compute various output patches, dramatically increasing the number of on-chip memory movements. This explains why a larger $G$ will significantly increase the practical efficiency, according to the experiment results in the paper.

2) *Computation latency $l_{compute}$.* The maximal FP32 computation throughput of the PE and the FLOPs required to compute an output tile are used to estimate the computation latency of each tile. The number of tiles and PEs can be used to determine the overall computation latency.

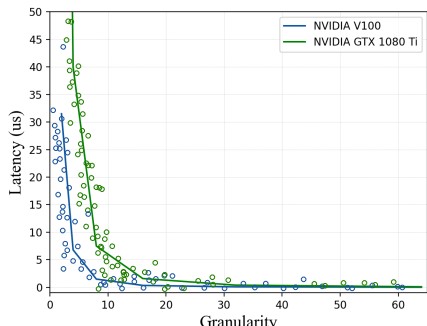

**Training-free theoretical model.** The latency prediction model is a training-free theoretical model suitable for general-purpose hardware, GPU. Unlike other works [35, 36] that focus on computation amounts, we directly optimize the on-device speed of the model because the speed depends on the memory access cost and the degree of parallelism as well. At the same time, ours are more efficient than the common method, hardware profiling. Furthermore, this modeling can be generalized in other networks as the latency prediction results on the 1st block in ResNet-50 in Figure 8. In specific, all networks are transformed into fixed matrix operations on GPU platforms (General Matrix Multiply, GeMM). The latency prediction model evaluates the speed of matrix multiplication and the corresponding data movement, so it is applicable to general networks.

Figure 8: **Latency prediction results of ResNet-50.** Results are tested on NVIDIA V100 and GTX 1080 Ti.

### A.2 Convolutional Modulation

Following [37], we replace the self-attention inside the transformer layer with a convolutional modulation layer. As shown in Figure 5(c), we modulate the value V with convolutional features. Let $X \in \mathbb{R}^{H \times W \times C}$ be input tokens, and we use depthwise convolution with kernel size $k \times k$ and the Hadamard product to calculate the output:

$$
\begin{aligned}
Z &= A \odot V, \\
A &= DConv_{k \times k}(W_1 X), \\
V &= W_2 X,
\end{aligned}
\tag{3}
$$

where $\odot$ is the Hadamard product, $W_1$ and $W_2$ are the weight matrics of two linear layers, and $DConv_{k \times k}$ denotes the depthwise convolution. The linear layers can be used to achieve the information interaction between channels. The weighted sum of all the pixels in the square area is the output for each spatial location. Our methods use convolution instead of self-attention to create associations, which are more memory-efficient (linear memory complexity), especially when processing high-resolution images. Due to the modulation operation, our method differs from traditional residual blocks and can adapt to the input content.

### A.3 BN-Based Swin-Transformer

We modify the basic structure of Swin-Transformer, WMSA/SWMSA, into WMSA$_{bn}$/SWMSA$_{bn}$ as shown in Figure 9. Compared to the original design with LN-Linear, a 13%~21% speedup is harvested with negligible accuracy degradation (<0.3%) on small models. In this paper, we use WMSA$_{bn}$/SWMSA$_{bn}$ as our design candidates.

### A.4 RepCNN for Branch Fusion

In Section 3.1.2, we apply RepCNN as a network component, fusing multiple branches into more single-branch substructures during inference. The detailed structure of RepCNN is illustrated in Figure 10.

### A.5 The Hardware-Oriented Decoder Structure

In figure 11 we present our decoder module, which consists of six layers. To improve efficiency in self-attention computation, we replace the channel-wise attention with convolution modulation in the first three layers. This modification allows for more efficient processing while still capturing important local dependencies. In the remaining three layers, we employ channel-wise attention, as

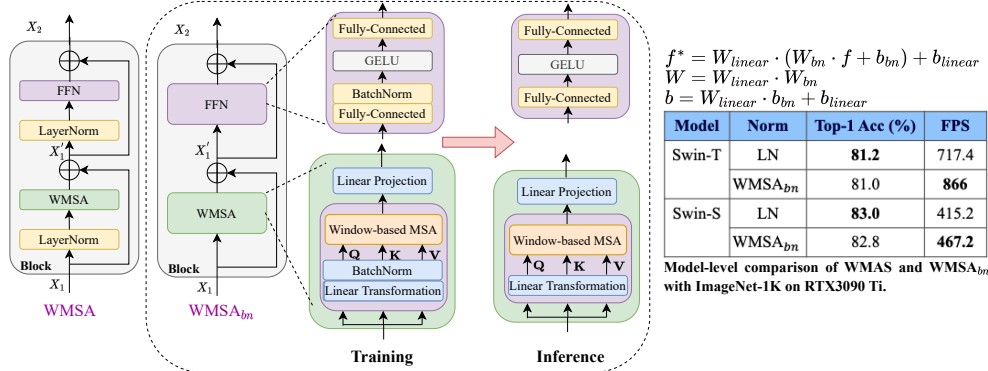

$$f^* = W_{linear} \cdot (W_{bn} \cdot f + b_{bn}) + b_{linear}$$
$$W = W_{linear} \cdot W_{bn}$$
$$b = W_{linear} \cdot b_{bn} + b_{linear}$$

| Model | Norm | Top-1 Acc (%) | FPS |
|-------|------|---------------|-----|
| Swin-T | LN | **81.2** | 717.4 |
| | WMSA$_{bn}$ | 81.0 | **866** |
| Swin-S | LN | **83.0** | 415.2 |
| | WMSA$_{bn}$ | 82.8 | **467.2** |

**Model-level comparison of WMAS and WMSA$_{bn}$ with ImageNet-1K on RTX3090 Ti.**

Figure 9: **WMSA$_{bn}$/SWMSA$_{bn}$ structure**. 13%~21% speedup can be achieved with <0.3 accuracy degradation on ImageNet-1K.

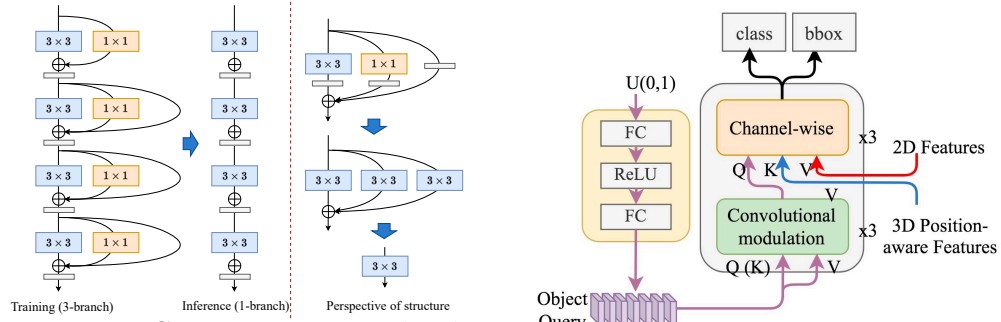

Figure 10: **RepCNN structure**. We also show the training status and the inference status of this structure, respectively.

Figure 11: **The Hardware-Oriented Decoder Structure.**

proposed in [38], as the global attention layer. This choice enables our model to effectively capture and incorporate global information into the decoding process, enhancing its overall performance.

Following [8], to ease the convergence difficulty in 3D scenes, we initialize a set of learnable anchor points in 3D world space with uniform distribution from 0 to 1. The coordinates of 3D anchor points are input to the query generator to generate initial object queries. The query generator is composed of two linear layers and one ReLU. The initial object queries will first go through the convolution modulation layers and then interact with the 3D position-aware features in the channel-wise attention layers. The updated object queries are further used to predict the object class and the 3D bounding boxes. Specifically, 2D image features are the value of the global attention blocks to represent the content of the image, and the 3D Position-aware Features are the key of the global attention blocks for a feasible positioning.

## A.6 Overall Framework

As shown in Figure 12, Given images $I=\{I^i \in \mathbb{R}^{3 \times H \times W}, i = 1, ..., N\}$ from $N$ views, the images are sent to the hardware-oriented backbone (HOB) to extract 2D multi-view features, $F^{2d}=\{F_i^{2d} \in \mathbb{R}^{C \times H \times W}, i = 1, \ldots, N\}$. The 3D coordinates are generated from camera frustum space and the coordinates of the previous frame $t - 1$ are aligned into the coordinate system of the current frame $t$ through the temporal aligned module (TAM). Then the 2D features and the 3D coordinates of the adjacent frames are concatenated together, respectively, and are forwarded to the 3D *position encoder* to generate the 3D-aware features. After that, the 3D-aware features are employed as our hardware-oriented decoder's key and value components. Further, detection queries, initialized from learnable 3D anchor points [39], are fed into the decoder and interact with the 3D-aware features. Lastly, the updated queries are input to the detection head for final prediction. We also deploy a latency-aware model slimming method strategy to generate efficient models for the target devices. The *coordinates generator* and 3D *position encoder* are built on PETR [8].

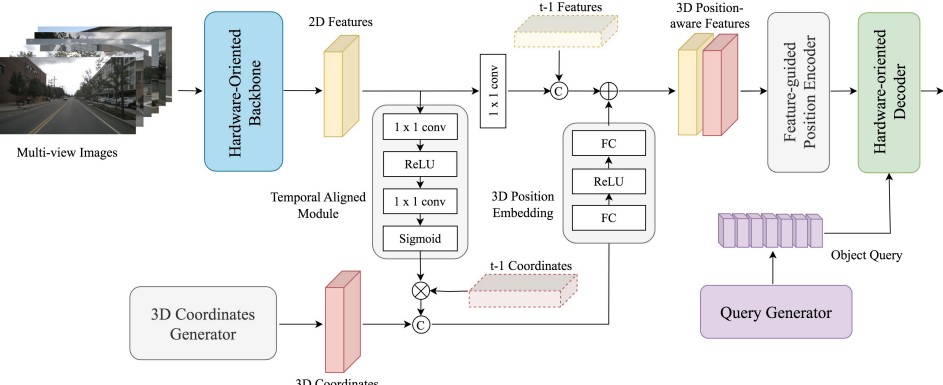

Figure 12: **The architecture of the proposed HotBEV paradigm**. The multi-view images go through our proposed hardware-oriented backbone to extract 2D features. The 3D coordinates first go through a 3D position embedding layer and then add together with the 2D features to obtain the 3D position-aware features. The object query can directly perceive the 3D object information by interacting with the 3D position-aware features in our proposed hardware-oriented decoder layer. $t-1$ time frame features along with a temporally aligned module are implemented to enhance the model estimation of velocity and attitude as well as robustness improvement.

### A.7 Implementation on Diverse General Vision Tasks

**Generalization ability on image classification.** To validate the generalization of our method beyond the limited scope of Bird's Eye View (BEV) modifications, we search for a new structure using NAS specifically for image classification tasks, leveraging our proposed hardware design paradigm, vision modeling logic, and efficiency-oriented operations such as convolution modulation and normalization fusion. We obtained a novel structure called HOBformer. In Table 7, we compared the performance of our new model with existing state-of-the-art classification models. This evaluation aimed to demonstrate that our approach is not only effective for BEV-related tasks but also applicable and competitive in the broader context of image classification. Specifically, our HOBformer$_{tiny}$ model outperforms EfficientNet-B0 by 2.0% in Top-1 accuracy, while simultaneously achieving a 4.3us reduction in latency. In comparison to MobileNetV2×1.0, our model achieves an 8.3% improvement in accuracy with a marginal increase of only 1.3us in latency.

Table 7: Comparison of different models on the ImageNet-1K. FPS is tested on V100 with FP32.

| Model | Params(M) | GMACs | Epochs | Top-1(%) | V100 Latency (us) |
|---|---|---|---|---|---|
| MobileNetV2×1.0 | 3.5 | 0.3 | 300 | 71.8 | 5.5 |
| ResNet50 | 25.5 | 4.1 | 300 | 78.5 | 10 |
| EfficientNet-B0 | 5.3 | 0.4 | 350 | 77.1 | 11.1 |
| DeiT-T | 5.9 | 1.2 | 300 | 74.5 | 7.8 |
| Swin-T | 29 | 4.5 | 300 | 81.3 | 24.2 |
| MobileViT-XS | 2.3 | 0.7 | 300 | 74.8 | 12.8 |
| HOBformer$_{tiny}$ | 12.4 | 1.5 | 300 | 79.1 | 6.8 |
| HOBformer$_{small}$ | 31 | 3.8 | 300 | 82.3 | 14.2 |

**Fusing batch normalization and convolutional modulation on existing architectures.** We integrate our proposed techniques fusing batch normalization and convolutional modulation into existing architectures. As shown in Table 8, for DETR-R50, the fusion of Batch Normalization (BN) in its backbone results in a speed improvement of 1.6 FPS. However, when we attempt to replace Layernorm with Batchnorm in the encoder and decoder sections and then fuse BN, the model's speed can be further increased by 0.3 and 0.5 FPS, respectively. Unfortunately, these modifications pose challenges in achieving convergence during model training.

Similarly, in deformable DETR, the ResNet backbone already incorporates BN fusion, yet the model fails to converge when the fusing technique is applied to its encoder and decoder. It becomes evident that directly replacing Layernorm with Batchnorm in the original self-attention is difficult, as it hampers model convergence. However, our design showcases one advantage of the window-based self-attention in terms of efficient implementation compared to other transformer architectures: it

Table 8: Integrating the proposed techniques into the 2D detection baselines (on V100).

| Method | Fusing batch normalization | Convolutional modulation | AP(%) | FPS |
|--------|---------------------------|-------------------------|-------|-----|
| DETR-R50 | N | N | 42 | 24 |
| DETR-R50 | Y (ResNet) | N | 42 | 25.6 |
| DETR-R50 | Y (ResNet + Encoder) | N | Not Converge | 25.9 |
| DETR-R50 | Y (ResNet + Decoder) | N | Not Converge | 26.1 |
| DETR-R50 | N | Y | 42.5 | 24.4 |
| DETR-R50 | Y (ResNet) | Y | 42.5 | 26 |
| deformable | Y (ResNet) | N | 43.9 | 19.1 |
| deformable | Y (ResNet + Encoder) | N | Not Converge | 19.4 |
| deformable | Y (ResNet + Encoder) | N | Not Converge | 19.6 |
| deformable | Y (ResNet) | Y | 44.3 | 19.6 |

allows for the replacement of Layernorm with Batchnorm and subsequent fusion with acceptable accuracy, as shown in Figure 9. Additionally, by leveraging convolutional modulation to model global attention, improvements in both speed and detection performance are observed. Specifically, on DETR-R50, there is a 0.5% increase in Average Precision (AP) and a gain of 0.4 FPS. Similarly, in deformable DETR, Convolutional Modulation leads to a 0.4% improvement in AP and an increase of 0.5 FPS.

## A.8 Additional Ablation Analysis

**Backbone performance on image classification and 2D object detection.** We have designed the hardware-oriented detector backbone(HOB) as our image backbone (Figure 5). Table 9 lists its results on the ImageNet classification and the computation complexity of the backbone. In Table 10, we compare the performance with two existing backbones: ResNet-50 and Swin-T on object detection. When integrated with ResNet-50 as the backbone, the model can speed up by 1.4 FPS but with 5.7 AP drops. With Swin-T as the backbone, the speed will decrease by 3.1 FPS with a 3.6 AP drop.

Table 9: Integrating the proposed techniques into the 2D detection baselines (on V100).

| Model | Top-1 Acc.(%) | AP(%) | GFLOPS of backbone | Overall GFLOPS |
|-------|---------------|-------|--------------------|----------------|
| HOTDETR$_{tiny}$ | 81.5 | 44.5 | 47 | 67 |
| HOTDETR$_{small}$ | 82.1 | 45.3 | 50 | 69 |
| HOTDETR$_{medium}$ | 82.7 | 46.4 | 56 | 73 |
| HOTDETR$_{large}$ | 83.2 | 46.8 | 92 | 108 |

Table 10: Integrating the proposed techniques into the 2D detection baselines (on V100).

| Model | Backbone | AP(%) | GFLOPS | FPS |
|-------|----------|-------|--------|-----|
| HOTDETR$_{tiny}$ | ResNet-50 | 38.8 | 62 | 27.6 |
| HOTDETR$_{tiny}$ | Swin-T | 40.9 | 71 | 23.1 |
| HOTDETR$_{tiny}$ | HOB$_{tiny}$ | 44.5 | 67 | 26.2 |

**Image embedding module.** Our major conclusions and speed analysis can be found in Section 1 and Figure 3. Here we illustrate more ablation studies for different settings. All the experiments are conducted on the HotBEV-nano. Compared to non-overlap large-kernel patch embedding, the proposed Image Embedding Module in HotBEV-nano achieves significant inference latency reduction by 44%~50% on multiple devices, while providing 2.8% higher accuracy on ImageNet-1K dataset as shown in Table 11. We demonstrate that convolution stem [40] can enhance model convergence and accuracy, and boost inference speed on multiple devices by a large margin as well, thus can be a good choice for non-overlapping patch embedding implementations.

**Study on the replacement of RepCNN in the HOB backbone.** To analyze the trade-off between the detection precision and the efficiency of the RepCNN utilization, we replace the WMSA$_{bn}$/SWMSA$_{bn}$ with the RepCNN in the $1st$ and $2nd$ stage. We experimentally find that detection performance is improved by 0.3%~0.5% AP after replacing WMSA with RepCNN in $S_1$ of our backbone (Table 12). The precision is only improved by 0.1% AP after replacing with RepCNN layers in the whole $S_1, S_2$. So a dedicated design is required here to extract the texture-level information effectively.

Table 11: The ablation analysis of Image Embedding Module (IEM) on ImageNet-1K. The speed results are test on $Latency_1$ for V100, $Latency_2$ for RTX2080 Ti, $Latency_3$ for GTX1080 Ti.

| IEM | Top-1 | $Latency_1$ | $Latency_2$ | $Latency_3$ |
|---|---|---|---|---|
| - | 79.6 | 0.059ms | 0.074ms | 0.64ms |
| ✓ | **82.4** | **0.033ms** | **0.041ms** | **0.032ms** |

Table 12: The ablation studies of replacement of RepCNN in the backbone. Note that $S_{11}$ indicates the $1st$ block of the $S_1$, and ✓ means we use the RepCNN block. The speed results are tested on one V100.

| $S_{11}$ | $S_{12}$ | $S_{21}$ | $S_{22}$ | NDS↑ | mAP↑ | FPS↑ |
|---|---|---|---|---|---|---|
| - | - | - | - | 0.372 | 0.320 | 11.8 |
| ✓ | - | - | - | 0.376 | 0.323 | 12.8 |
| ✓ | ✓ | - | - | **0.381** | **0.325** | 13.7 |
| ✓ | ✓ | ✓ | - | 0.379 | 0.324 | 14.5 |
| ✓ | ✓ | ✓ | ✓ | 0.373 | 0.321 | **14.7** |

**Temporal modeling.** We analyze the effect of temporal modeling, which consists of two parts: 3D coordinates alignment (CA) and data-aware strategy (DA). As shown in Table 13, the performance is improved by 2.8% NDS and 1.2% mAP with CA. The mAVE metric is 0.94 m/s, which shows an 8% improvement to the baseline. The mATE, mASE, and mAOE, which describe the form and attitude of the object, are improved 0.3%~2.1%. After assembling with DA, the NDS metric is increased by 1%, and mAP is increased by 0.5%. It illustrates that DA can further enhance the temporal alignment by adding prior information, e.g., depth information, or reducing the perturbation of the external parameter.

Table 13: Ablation studies of two components in the TAM.

| CA | DA | NDS↑ | mAP↑ | mATE↓ | mASE↓ | mAOE↓ | mAVE↓ | mAAE↓ |
|---|---|---|---|---|---|---|---|---|
| - | - | 0.350 | 0.309 | 0.780 | 0.278 | 0.570 | 1.12 | 0.215 |
| ✓ | - | 0.378 | 0.324 | 0.759 | 0.275 | 0.550 | 0.940 | 0.213 |
| ✓ | ✓ | **0.388** | **0.329** | **0.748** | **0.273** | **0.545** | **0.930** | **0.208** |

### A.9 Robustness Analysis

In realistic scenarios, various unexpected situations are encountered. Therefore, it is particularly important to improve the robustness of the camera system. Some works have explored this direction. LSS considers extraneous noise and camera dropouts during testing. BEVFormer shows temporal information can improve the system's robustness. Considering three potential error encounters, 1) Camera shake/offset caused by external forces. 2) Miss-capturing information/scene caused by camera dysfunction. 3) Camera time delay caused by long exposure time. Robustness can be evaluated with mATE, mASE, and mAOE scores. The temporal aligned module (Figure 12) introduces the 2D feature to the 3D feature. This data-aware approach can help improve the robustness. We evaluate the robustness in two ways:

**Add extrinsic noises.** We compare the performance of three models: No 3D coordinates alignment (CA) and data-aware strategy (DA) module, CA only module, CA, and DA module. Figure 13 shows the mAP and NDS score of each model under different degrees of extrinsic noises. Our temporal aligned module can improve the 3D detection of the model. Specifically, when Rmax=8, the accuracy drops between w. DA and w/o. DA is 3.9% vs. 4.4%. This shows that DA helps improve robustness as it can slow down the performance degradation when noise increases.

**Remove one of the six cameras.** As shown in Table 14, we can observe that removing the back (-12.5% NDS, -10.5% mAP) or front (-3.3% NDS, -3% mAP) camera has the greatest impact on accuracy. Our temporal-aligned module can also improve the robustness of the camera dysfunction scenario (e.g., 9.9% vs. 10.5% accuracy drop between w. DA and w/o. DA).

**Comparison with randomly searched models.** Besides the proposed basic hardware-efficient design, it is still important to determine appropriate model configurations, e.g., depth and width, to achieve promising performance. To illustrate the benefits of our Latency-aware Model Slimming Strategy,

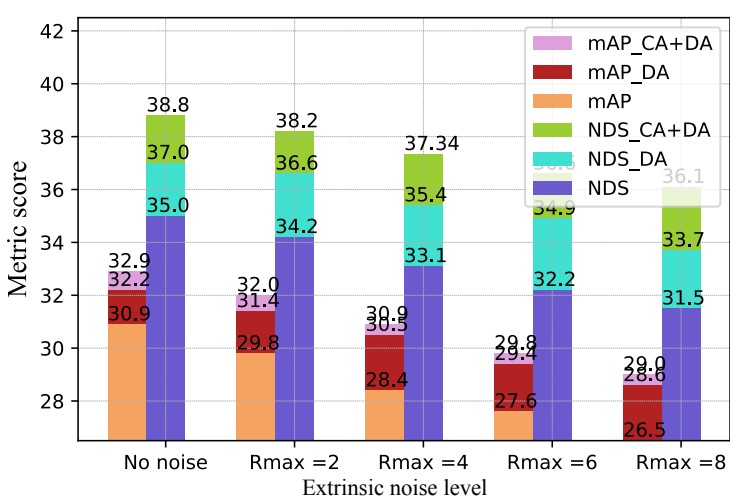

Figure 13: **Left**: Results on the nuScenes val set with extrinsic noises $R_{max}$. **Middle and right**: The trade-off between performance (NDS) and hardware efficiency (FPS) for different detection methods on the nuScenes val set with different GPUs.

| CA | DA | All | Front | Front Right | Front Left | Back | Back Right | Back Left |
|---|---|---|---|---|---|---|---|---|
| | | | | NDS | | | | |
| ✗ | ✗ | 35.0 | 31.7 | 33.7 | 33.7 | 22.5 | 32.6 | 33.1 |
| ✗ | ✓ | 37.0 | 33.6 | 35.6 | 35.6 | 24.9 | 35.1 | 35.8 |
| ✓ | ✓ | **38.8** | **33.8** | **35.8** | **35.7** | **25.5** | **35.6** | **36.1** |
| | | | | mAP | | | | |
| ✗ | ✗ | 30.9 | 27.9 | 29.1 | 29.2 | 20.4 | 27.6 | 28.3 |
| ✗ | ✓ | 32.2 | 29.8 | 30.9 | 31.0 | 23.3 | 30.6 | 31.0 |
| ✓ | ✓ | **32.9** | **29.9** | **31.1** | **31.1** | **23.5** | **30.7** | **31.2** |

Table 14: Results on the nuScenes val set when removing one camera each time.

we randomly sample networks from our search space that have the same mapping latency, i.e., 14.5 FPS, as our model HotBEV-nano. The sampled networks are denoted as Random$_1$~Random$_4$, which are either deeper and narrower or shallower and wider than HotBEV-nano. We train the sampled models on nuScenes with the same training recipe as HotBEV-nano. The comparison results of these models are shown in Table 15. Our searched HotBEV-nano has better latency or higher NDS/mAP on nuScenes than the randomly sampled networks.

Table 15: Analysis of Latency-aware Model Slimming. The FPS is obtained on V100.

| Model | FPS↑ | NDS↑ | mAP↑ |
|---|---|---|---|
| HotBEV-nano | **14.5** | **0.388** | **0.329** |
| Random$_1$ | 14.5 | 0.380 | 0.312 |
| Random$_2$ | 14.4 | 0.374 | 0.305 |
| Random$_3$ | 14.4 | 0.383 | 0.315 |
| Random$_4$ | 14.6 | 0.372 | 0.295 |

## A.10 Latency-aware Model Slimming

We provide the details of the proposed fast latency-aware model slimming strategy in Algorithm 1. Relative formulations can be found in Section 3.3. The proposed latency-aware model slimming strategy is speed-oriented for the target device, which does not need retraining for each sub-network. The importance score for each device-design choice is estimated based on the trainable architecture parameter $r$.

**Algorithm 1:** Latency-aware Model Slimming

---

**1** **Given**: speed look up table $F = \{RepCNN, WMSA_{bn}, SWMSA_{bn}\}_{dim=32\times}$,

**2** $\{RepCNN, WMSA_{bn}, SWMSA_{bn}, Channel-wise, Self-atten\}_{dim=32\times}$;

**3** **Requirement**: Final throughput budget: $\sum F \approx \top$;

**4** *Super-net Pretraining:*

**5** **foreach** *epoch* **do**

**6**     **foreach** *iteration* **do**

**7**         **foreach** $HP_{i,j}$ **do**

**8**             $\kappa_{i+1} = \sum_n \frac{e^{(r_i^n + \varepsilon_i^n)/\tau}}{\sum_n e^{(r_i^n + \varepsilon_i^n)/\tau}} \cdot HP_{i,j}(\kappa_i)$;

**9**         **end**

**10**         $\pounds = criterion\_loss\_function(output, label)$;

**11**         backpropagate ($\pounds$);

**12**         update parameters;

**13**     **end**

**14** **end**

**15** $\triangle$ Obtain the Supernet.

**16** *Speed-Driven Model thinning:*

**17** E $\in$ {Layer Reduction (LR), Width Reduction (WR), I Reduction (IR), WMSA Reduction (WMR), SWMSA Reduction (SR)};

**18** Calculate the importance of $HP_{i,j}$ through $\mathbb{M}_{i,j} = \frac{r_i^{RepCNN} + r_i^{WMSA} + r_i^{SWMSA}}{r_i^I}$ or $\frac{r_i^{WMSA} + r_i^{SWMSA}}{r_i^I}$;

**19** **while** $\sum F > \top$ **do**

**20**     $LR \leftarrow argmin_{\mathbb{M}_{i,j}}(HP_{i,j})$;

**21**     $IR \leftarrow argmin_{\sum_j \mathbb{M}_{i,j}}(HP_{i,j})$;

**22**     $SR \leftarrow argmin_{\sum_j \mathbb{M}_{i,j}}$;

**23**     $(HP_{i,j}), WMR \leftarrow argmin_{\sum_j \mathbb{M}_{i,j}}(HP_{i,j})$;

**24**     $WR \leftarrow argmin_{\sum_j \mathbb{M}_{i,j}}(HP_{i,j})$;

**25**     Conduct Evolution $= argmin_{\frac{AP_{drop}}{F_{i,j}}}(E)$;

**26** **end**

**27** $\triangle$ Obtain the Subnet with the target FPS.

**28** *Train the searched architecture from scratch:*

**29** SDG-Training method.

**30** $\triangle$ Obtain the final model.

---

## A.11 Visualization of Feature Map

We further visualize the feature maps from each backbone stage of HOB-nano in Figure 14. In the first and second stages, our HOB-nano can capture sufficient low-level semantic information to detect small objects. The third stage then focuses on medium and large objects. The last stage only responds to large objects. The observations demonstrate that our HOB can boost semantic information in low-level features and thus enhance detection precision.

## A.12 Model Configuration of HotBEV

The detailed network architectures for the backbone of HotBEV-nano, HotBEV-tiny, and HotBEV-base on multiple GPUs are provided in Table 16, Table 17, and Table 18. We report the resolution and number of blocks for each stage. In addition, the width of HotBEV is specified as the embedding dimension (Embed., Dim.). As for the MHSA block, the dimension of Query and Key is provided.

| Input Images | Stage 1 | Stage 2 | Stage 3 | Stage 4 |
|---|---|---|---|---|

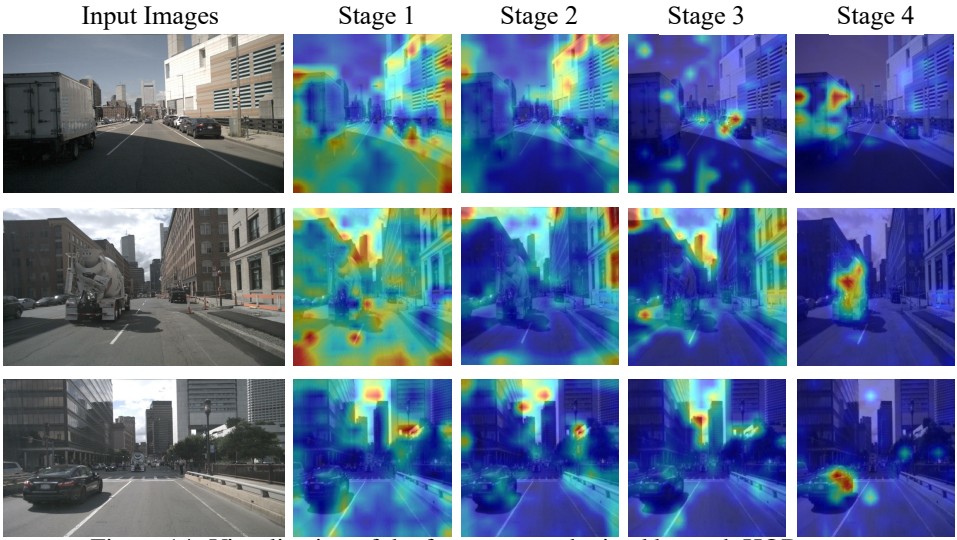

Figure 14: Visualization of the feature map obtained by each HOB stage.

Table 16: Detailed Architectures of the HOB backbone of HotBEV on V100. $D_{QK}$ is the dimension of Queries and Keys. $Exp$ refers to the expansion ratio of the MLP block.

| Stage | Resolution | Type | Config | | HotBEV Nano | HotBEV Tiny | HotBEV Base |
|---|---|---|---|---|---|---|---|
| Image Embed. | $\frac{H}{2} \times \frac{W}{2}$ | Image Embed. | Patch Size | | k=3x3,s=2 | | |
| | | | Embed. Dim. | | 48 | 64 | 64 |
| Image Embed. | $\frac{H}{4} \times \frac{W}{4}$ | Image Embed. | Patch Size | | k=3x3,s=2 | | |
| | | | Embed. Dim. | | 96 | 128 | 128 |
| 1 | $\frac{H}{4} \times \frac{W}{4}$ | Local Attention | RepCNN= | $\begin{matrix} Embed. & Kernel \\ Stride & Exp \end{matrix}$ | [96, 3, 1, 4]×1 | [128, 3, 1, 4] ×1 | [128, 3, 1, 4] ×1 |
| | | | SWMSA$_{bn}$= | $\begin{matrix} Embed. & D_{QK} \\ Heads & Exp \end{matrix}$ | [96, 96, 3, 4] ×1 | [128, 128, 4, 4] ×1 | [128, 128, 4, 4] ×1 |
| | | Global Position Generator | DWConv= | $\begin{matrix} Embed. & Kernel \\ Stride & Exp \end{matrix}$ | [96, 3, 1, 4] ×1 | [128, 3, 1, 4] ×1 | [128, 3, 1, 4] ×1 |
| | | Global Attention | Convolutional Modulation= | $\begin{matrix} Embed. & D_{QK} \\ Heads & Exp \end{matrix}$ | [96, 96, 3, 4] ×1 | [128,128, 4, 4] ×1 | [128, 128, 4, 4]×1 |
| 2 | $\frac{H}{8} \times \frac{W}{8}$ | Patch Embed. | Patch Size | | k=3X3, s=2 | | |
| | | | Embed. Dim. | | 96 | 128 | 128 |
| | | Local Attention | RepCNN= | $\begin{matrix} Embed. & Kernel \\ Stride & Exp \end{matrix}$ | [96, 3, 1, 4] ×1 | [128, 3, 1, 4] ×1 | [128, 3,1,4] ×1 |
| | | | SWMSA$_{bn}$= | $\begin{matrix} Embed. & D_{QK} \\ Heads & Exp \end{matrix}$ | [96, 96, 3, 4]×1 | [128, 128, 4, 4]×1 | [128, 128, 4, 4]×1 |
| | | Global Position Generator | DWConv= | $\begin{matrix} Embed. & Kernel \\ Stride & Exp \end{matrix}$ | [96, 3,1,4] ×1 | [128, 3,1,4] ×1 | 128, 3,1,4] ×1 |
| | | Global Attention | Convolutional Modulation= | $\begin{matrix} Embed. & D_{QK} \\ Heads & Exp \end{matrix}$ | [96, 96, 3, 4]×1 | [128, 128, 4, 4]×1 | [128, 128, 4, 4]×1 |
| | | SAM | Convolutional Modulation= | $\begin{matrix} Embed. & D_{QK} \\ Heads & Exp \end{matrix}$ | [96, 96, 3, 4] ×1 | [128, 128, 4, 4]×1 | [128, 128, 4, 4]×1 |
| 3 | $\frac{H}{16} \times \frac{W}{16}$ | Patch Embed. | Patch Size | | k=3X3, s=2 | | |
| | | | Embed. Dim. | | 192 | 256 | 224 |
| | | Local Attention | {RepCNN= | $\begin{matrix} Embed. & Kernel \\ Stride & Exp \end{matrix}$, | {[192, 3,1,4] , [192, 192, 6, 4]}×2 | {[256, 3,1,4] , [256, 256, 8, 4]}×2 | {[224, 3,1,4] , [224, 224, 7, 4]}×5 |
| | | | SWMSA$_{bn}$= | $\begin{matrix} Embed. & D_{QK} \\ Heads & Exp \end{matrix}$} | | | |
| | | | {WMSA$_{bn}$= | $\begin{matrix} Embed. & D_{QK} \\ Heads & Exp \end{matrix}$, | {[192, 192, 6, 4], [192, 192, 6, 4]}×1 | {[256, 256, 8, 4], [256, 256, 8, 4]}×1 | {[224, 224, 7, 4], [224, 224, 7, 4]}×4 |
| | | | SWMSA$_{bn}$= | $\begin{matrix} Embed. & D_{QK} \\ Heads & Exp \end{matrix}$} | | | |
| | | Global Position Generator | DWConv= | $\begin{matrix} Embed. & Kernel \\ Stride & Exp \end{matrix}$ | [192, 3,1,4] ×1 | [256, 3,1,4] ×1 | [224, 3,1,4] ×1 |
| | | Global Attention | Convolutional Modulation= | $\begin{matrix} Embed. & D_{QK} \\ Heads & Exp \end{matrix}$ | [192, 192, 6, 4]×1 | [256, 256, 8, 4]×1 | [224, 224, 7, 4]×1 |
| | | SAM | Convolutional Modulation= | $\begin{matrix} Embed. & D_{QK} \\ Heads & Exp \end{matrix}$ | [96, 96, 3, 4]×1 | [128, 128, 4, 4]×1 | [128, 128, 4, 4]×1 |
| 4 | $\frac{H}{32} \times \frac{W}{32}$ | Patch Embed. | Patch Size | | k=3X3, s=2 | | |
| | | | Embed. Dim. | | 288 | 384 | 384 |
| | | Local Attention | {WMSA$_{bn}$= | $\begin{matrix} Embed. & D_{QK} \\ Heads & Exp \end{matrix}$, | {[288, 288, 9, 4], [288, 288, 9, 4]}×1 | {[384, 384, 12, 4], [384, 384, 12, 4]}×1 | {[384, 384, 12, 4], [384, 384, 12, 4]}×1 |
| | | | SWMSA$_{bn}$= | $\begin{matrix} Embed. & D_{QK} \\ Heads & Exp \end{matrix}$} | | | |
| | | Global Position Generator | DWConv= | $\begin{matrix} Embed. & Kernel \\ Stride & Exp \end{matrix}$ | [288, 3,1,4] ×1 | [384, 3,1,4] × 1 | [384, 3,1,4] ×1 |
| | | Global Attention | Convolutional Modulation= | $\begin{matrix} Embed. & D_{QK} \\ Heads & Exp \end{matrix}$ | [288, 288, 9, 4]×1 | [384, 384, 12, 4]×1 | [384, 384, 12, 4]×1 |
| | | SAM | Convolutional Modulation= | $\begin{matrix} Embed. & D_{QK} \\ Heads & Exp \end{matrix}$ | [192, 192, 6, 4]×1 | [256, 256, 8, 4]×1 | [224, 224, 7, 4]×1 |

Table 17: Detailed Architectures of the HOB backbone of HotBEV on GTX 2080 ti. $D_{QK}$ is the dimension of Queries and Keys. $Exp$ refers to the expansion ratio of the MLP block.

| Stage | Resolution | Type | Config | | HotBEV Nano | Tiny | Base |
|---|---|---|---|---|---|---|---|
| Image Embed. | $\frac{H}{2}\times\frac{W}{2}$ | Image Embed. | Patch Size | | k=3x3,s=2 | | |
| | | | Embed. Dim. | | 48 | 64 | 64 |
| | $\frac{H}{4}\times\frac{W}{4}$ | Image Embed. | Patch Size | | k=3x3,s=2 | | |
| | | | Embed. Dim. | | 96 | 128 | 128 |
| 1 | $\frac{H}{4}\times\frac{W}{4}$ | Local Attention | {RepCNN=[Embed./Stride, Kernel/Exp], SWMSA$_{bn}$=[Embed./Heads, $D_{QK}$/Exp]} | | [[96, 3, 1, 4], [96, 96, 3, 4]]×2 | [[128, 3,1,4], [128, 128, 4, 4]]×2 | [[128, 3,1,4], [128, 128, 4, 4]]×5 |
| | | | {WMSA$_{bn}$=[Embed./Heads, $D_{QK}$/Exp], SWMSA$_{bn}$=[Embed./Heads, $D_{QK}$/Exp]} | | [[96, 96, 3, 4], [96, 96, 3, 4]]×1 | [[128, 128, 4, 4], [128, 128, 4, 4]]×1 | [[128, 128, 4, 4], [128, 128, 4, 4]]×4 |
| | | Global Position Generator | DWConv=[Embed./Stride, Kernel/Exp] | | [96, 3, 1, 4] ×1 | [128, 3, 1, 4] ×1 | [128, 3, 1, 4] ×1 |
| | | Global Attention | Convolutional Modulation=[Embed./Heads, $D_{QK}$/Exp] | | [96, 96, 3, 4] ×1 | [128,128, 4, 4] ×1 | [128, 128, 4, 4]×1 |
| 2 | $\frac{H}{8}\times\frac{W}{8}$ | Patch Embed. | Patch Size | | k=3X3, s=2 | | |
| | | | Embed. Dim. | | 96 | 128 | 128 |
| | | Local Attention | RepCNN=[Embed./Stride, Kernel/Exp] | | [96, 3, 1, 4] ×1 | [128, 3, 1, 4] ×1 | [128, 3,1,4] ×1 |
| | | | SWMSA$_{bn}$=[Embed./Heads, $D_{QK}$/Exp] | | [96, 96, 3, 4] ×1 | [128, 128, 4, 4]×1 | [128, 128, 4, 4]×1 |
| | | Global Position Generator | DWConv=[Embed./Stride, Kernel/Exp] | | [96, 3,1,4] ×1 | [128, 3,1,4] ×1 | 128, 3,1,4 ×1 |
| | | Global Attention | Convolutional Modulation=[Embed./Heads, $D_{QK}$/Exp] | | [96, 96, 3, 4]×1 | [128, 128, 4, 4]×1 | [128, 128, 4, 4]×1 |
| | | SAM | Convolutional Modulation=[Embed./Heads, $D_{QK}$/Exp] | | [96, 96, 3, 4] ×1 | [128, 128, 4, 4]×1 | [128, 128, 4, 4]×1 |
| 3 | $\frac{H}{16}\times\frac{W}{16}$ | Patch Embed. | Patch Size | | k=3X3, s=2 | | |
| | | | Embed. Dim. | | 192 | 256 | 224 |
| | | Local Attention | {RepCNN=[Embed./Stride, Kernel/Exp], SWMSA$_{bn}$=[Embed./Heads, $D_{QK}$/Exp]} | | [[192, 3,1,4], [192, 192, 6, 4]]×2 | [[256, 3,1,4], [256, 256, 8, 4]]×2 | [[224, 3,1,4], [224, 224, 7, 4]]×5 |
| | | | {WMSA$_{bn}$=[Embed./Heads, $D_{QK}$/Exp], SWMSA$_{bn}$=[Embed./Heads, $D_{QK}$/Exp]} | | [[192, 192, 6, 4], [192, 192, 6, 4]]×1 | [[256, 256, 8, 4], [256, 256, 8, 4]]×1 | [[224, 224, 7, 4], [224, 224, 7, 4]]×4 |
| | | Global Position Generator | DWConv=[Embed./Stride, Kernel/Exp] | | [192, 3,1,4] ×1 | [256, 3,1,4] ×1 | [224, 3,1,4] ×1 |
| | | Global Attention | Convolutional Modulation=[Embed./Heads, $D_{QK}$/Exp] | | [192, 192, 6, 4]×1 | [256, 256, 8, 4]×1 | [224, 224, 7, 4]×1 |
| | | SAM | Convolutional Modulation=[Embed./Heads, $D_{QK}$/Exp] | | [96, 96, 3, 4]×1 | [128, 128, 4, 4]×1 | [128, 128, 4, 4]×1 |
| 4 | $\frac{H}{32}\times\frac{W}{32}$ | Patch Embed. | Patch Size | | k=3X3, s=2 | | |
| | | | Embed. Dim. | | 288 | 384 | 384 |
| | | Local Attention | {WMSA$_{bn}$=[Embed./Heads, $D_{QK}$/Exp], SWMSA$_{bn}$=[Embed./Heads, $D_{QK}$/Exp]} | | [[288, 288, 9, 4], [288, 288, 9, 4]]×1 | [[384, 384, 12, 4], [384, 384, 12, 4]]×1 | [[384, 384, 12, 4], [384, 384, 12, 4]]×1 |
| | | Global Position Generator | DWConv=[Embed./Stride, Kernel/Exp] | | [288, 3,1,4] ×1 | [384, 3,1,4] × 1 | [384, 3,1,4] ×1 |
| | | Global Attention | Convolutional Modulation=[Embed./Heads, $D_{QK}$/Exp] | | [288, 288, 9, 4]×1 | [384, 384, 12, 4]×1 | [384, 384, 12, 4]×1 |
| | | SAM | Convolutional Modulation=[Embed./Heads, $D_{QK}$/Exp] | | [192, 192, 6, 4]×1 | [256, 256, 8, 4]×1 | [224, 224, 7, 4]×1 |

Table 18: Detailed Architectures of the HOB backbone of HotBEV on GTX 1080 ti. $D_{QK}$ is the dimension of Queries and Keys. $Exp$ refers to the expansion ratio of the MLP block.

| Stage | Resolution | Type | Config | | HotBEV Nano | Tiny | Base |
|---|---|---|---|---|---|---|---|
| Image Embed. | $\frac{H}{2}\times\frac{W}{2}$ | Image Embed. | Patch Size | | k=3x3,s=2 | | |
| | | | Embed. Dim. | | 48 | 64 | 64 |
| | $\frac{H}{4}\times\frac{W}{4}$ | Image Embed. | Patch Size | | k=3x3,s=2 | | |
| | | | Embed. Dim. | | 96 | 128 | 128 |
| 1 | $\frac{H}{4}\times\frac{W}{4}$ | Local Attention | RepCNN=[Embed./Stride, Kernel/Exp] | | [96, 3, 1, 4]×1 | [128, 3, 1, 4] ×1 | [128, 3, 1, 4] ×1 |
| | | | SWMSA$_{bn}$=[Embed./Heads, $D_{QK}$/Exp] | | [96, 96, 3, 4] ×1 | [128, 128, 4, 4] ×1 | [128, 128, 4, 4] ×1 |
| | | Global Position Generator | DWConv=[Embed./Stride, Kernel/Exp] | | [96, 3, 1, 4] ×1 | [128, 3, 1, 4] ×1 | [128, 3, 1, 4] ×1 |
| | | Global Attention | Convolutional Modulation=[Embed./Heads, $D_{QK}$/Exp] | | [96, 96, 3, 4] ×1 | [128,128, 4, 4] ×1 | [128, 128, 4, 4]×1 |
| 2 | $\frac{H}{8}\times\frac{W}{8}$ | Patch Embed. | Patch Size | | k=3X3, s=2 | | |
| | | | Embed. Dim. | | 96 | 128 | 128 |
| | | Local Attention | RepCNN=[Embed./Stride, Kernel/Exp] | | [96, 3, 1, 4] ×1 | [128, 3, 1, 4] ×1 | [128, 3,1,4] ×1 |
| | | | SWMSA$_{bn}$=[Embed./Heads, $D_{QK}$/Exp] | | [96, 96, 3, 4]×1 | [128, 128, 4, 4]×1 | [128, 128, 4, 4]×1 |
| | | Global Position Generator | DWConv=[Embed./Stride, Kernel/Exp] | | [96, 3,1,4] ×1 | [128, 3,1,4] ×1 | 128, 3,1,4 ×1 |
| | | Global Attention | Convolutional Modulation=[Embed./Heads, $D_{QK}$/Exp] | | [96, 96, 3, 4]×1 | [128, 128, 4, 4]×1 | [128, 128, 4, 4]×1 |
| | | SAM | Convolutional Modulation=[Embed./Heads, $D_{QK}$/Exp] | | [96, 96, 3, 4] ×1 | [128, 128, 4, 4]×1 | [128, 128, 4, 4]×1 |
| 3 | $\frac{H}{16}\times\frac{W}{16}$ | Patch Embed. | Patch Size | | k=3X3, s=2 | | |
| | | | Embed. Dim. | | 192 | 256 | 224 |
| | | Local Attention | {WMSA$_{bn}$=[Embed./Heads, $D_{QK}$/Exp], SWMSA$_{bn}$=[Embed./Heads, $D_{QK}$/Exp]} | | [[192, 3,1,4], [192, 192,6,4]]×1 | [[256, 256, 8, 4], [256, 256, 8, 4]]×1 | [[256, 256, 8, 4], [256, 256, 8, 4]]×1 |
| | | Global Position Generator | DWConv=[Embed./Stride, Kernel/Exp] | | [192, 3,1,4] ×1 | [256, 3,1,4] ×1 | [224, 3,1,4] ×1 |
| | | Global Attention | Convolutional Modulation=[Embed./Heads, $D_{QK}$/Exp] | | [192, 192, 6, 4]×1 | [256, 256, 8, 4]×1 | [224, 224, 7, 4]×1 |
| | | SAM | Convolutional Modulation=[Embed./Heads, $D_{QK}$/Exp] | | [96, 96, 3, 4]×1 | [128, 128, 4, 4]×1 | [128, 128, 4, 4]×1 |
| 4 | $\frac{H}{32}\times\frac{W}{32}$ | Patch Embed. | Patch Size | | k=3X3, s=2 | | |
| | | | Embed. Dim. | | 288 | 384 | 384 |
| | | Local Attention | {RepCNN=[Embed./Stride, Kernel/Exp], SWMSA$_{bn}$=[Embed./Heads, $D_{QK}$/Exp]} | | [[288, 3,1,4], [288, 288, 9, 4]]×2 | [[384, 3,1,4], [384, 384, 12, 4]]×2 | [[384, 3,1,4], [384, 384, 12, 4]]×5 |
| | | | {WMSA$_{bn}$=[Embed./Heads, $D_{QK}$/Exp], SWMSA$_{bn}$=[Embed./Heads, $D_{QK}$/Exp]} | | [[288, 288, 9, 4], [288, 288, 9, 4]]×1 | [[384, 384, 12, 4], [384, 384, 12, 4]]×1 | [[384, 384, 12, 4], [384, 384, 12, 4]]×4 |
| | | Global Position Generator | DWConv=[Embed./Stride, Kernel/Exp] | | [288, 3,1,4] ×1 | [384, 3,1,4] × 1 | [384, 3,1,4] ×1 |
| | | Global Attention | Convolutional Modulation=[Embed./Heads, $D_{QK}$/Exp] | | [288, 288, 9, 4]×1 | [384, 384, 12, 4]×1 | [384, 384, 12, 4]×1 |
| | | SAM | Convolutional Modulation=[Embed./Heads, $D_{QK}$/Exp] | | [192, 192, 6, 4]×1 | [256, 256, 8, 4]×1 | [224, 224, 7, 4]×1 |