# OpenReview forum: "HotBEV: Hardware-oriented Transformer-based Multi-View 3D Detector for BEV Perception"
_NeurIPS.cc/2023/Conference — NeurIPS 2023 poster_

### Official Review · Reviewer_Mv7q · 2023-07-03

**Soundness:** 3 good
**Presentation:** 3 good
**Contribution:** 3 good
**Rating:** 6
**Confidence:** 3

**Summary:**

This submission introduces a carefully-crafted transformer model family (models with varying speed-accuracy trade-off) for 3D detection from multi-view camera data. The adopted model design methodology prioritises hardware-efficiency, customising the proposed architecture to the target GPU. For this purpose, an analytical performance model is developed and queried to estimate the inference latency and guide different design choices. Additionally, following a module-level benchmarking, targeted architectural changes are introduced to alleviate the main identified computational bottlenecks in the backbone, normalisation and fusion layer design. The proposed model is trained in the form of a super-network, from which targeted sub-models can be extracted, offering a speed-accuracy trade-off that dominates the pareto frontier between the examined baselines.

**Strengths:**

- The aim of the paper to design efficient 3D detection models that are able to achieve realistic latency gains, rather than theoretical workload reduction, is very important and well-motivated.
- Additionally, offering a design methodology that is able to customise the model design to the target computation platform facilitates adaptability and is of great benefit to the community.
- The results of the conducted benchmarking are insightful and offer detailed information for improving inference efficiency on similar tasks.
- The proposed approach dominates the speed-accuracy optimality frontier on 3D detection from BEV data, compared to widely adopted baselines.

**Weaknesses:**

- The design choice to focus on multi-frame models instead of those adopting 3D representations is not sufficiently justified, nor backed by a corresponding discussion back by experimental evidence. Particularly since this design choice restricts the re-use of the proposed approach to data from other sensors, such as LiDAR.
- The examined GPU devices are on the power-hungry end of the spectrum. For self-driving scenarios, it also makes sense to consider embedded GPU devices too.
-The writing of the paper becomes a bit dense and hard to follow in certain sections of the methodology, leaving some unclear points.

**Questions:**

- Please provide further discussion/results on how the proposed methods speed and accuracy compares with 3D reconstruction based BEV detectors.
- How would the proposed methodology perform when targeting embedded GPUs, such as AGX Xavier or Orin.
- The proposed latency modelling seems to study memory and computation in isolation. In the case of memory-bounded layers, does the proposed model capture the latency-penalty of PEs who are stalling due to data starvation (e.g. similar to Roofline model analysis) ?
- Would an oracle/benchmarking-based approach, considering the actual latency of each architecture result in different design choices to those taken considering the estimation of the developed model ?

Typos:
- In Table 3 the HW properties of GTX2080 and GTX1080 appear to be identical. Is this correct ?
- Line 213: [3] -> Figure [3]

**Limitations:**

The manuscript briefly discusses some limitations of the current approach, mostly in the form of future/active work undertaken by the authors. A more detailed discussion of the assumptions and corresponding limitations of the approach would add value to the manuscript.

---

> ### Author Rebuttal · Authors · 2023-08-10
>
> **ReW1:**  Thank you for this question to give us a chance to clarify our motivations.
> Currently, the pre-fusion family, such as BevFusion, is very expensive to deploy in practice. The current in-vehicle market is dominated by camera-only solutions, such as Tesla, or post-fusion of camera and lidar signals, such as XPeng Motors, whose camera part still deploys camera-only solutions [1].
> Hence, our research goal is to explore the efficient and effective camera-only 3D detection series for practical implementation. So we choose the state-of-the-art frameworks, multi-frame models as the starting point of our research.
> Moreover, according to [2], the camera processing modules occupy 33% ~ 42% of the whole latency distribution for the post-fusion system. Our hardware-oriented design with enhanced image feature representation can be leveraged in the camera encoder part of the current in-vehicle detection system to improve on-device efficiency.
>
> **ReW2:**  Thanks for your valuable suggestions. It is necessary to test the speed on an actual commercial chip. As shown in Table E, we test our HotBEV models on Orin to validate our framework. Before testing, we quantized our model into INT8 with a tensorRT engine. And then run the test 50 times for each model in order to obtain stable results.
>
> Table E. Results on Orin.
> |Methods | Backbone | Resolution| NDS↑ |mAP↑ | FPS|
> |------------|-----|-----|-----|-----|-----|
> |HotBEV	|HOB-nano	|512x1408	|0.47	|0.385	|31.8|
> |HotBEV	|HOB-tiny	|512x1408	|0.512	|0.407	|20.4|
> |HotBEV	|HOB-base	|512x1408	|0.525	|0.427	|16.1|
>
> **ReW3:** Thanks for your valuable suggestions. We will put more related works and experiments in the appendix part. We will focus on methodology in the main body of our final version.
>
> **ReQ1:** Thank you for guiding us to make a more comprehensive comparison with other branch methods. Table F shows that 3D reconstruction-based BEV detectors are better than our methods by up to 19.8 NDS but with inferior on-device speed than ours by up to 19.4 FPS. Even though the detection performance of image-only detectors cannot be superior to Lidar-only methods, these results also provide us motivation to implement our design into one of the commercial mainstream, post-fusion systems to improve hardware efficiency, as mentioned in W1.
>
>
> Table F. Comparison with 3D reconstruction-based BEV detectors.
> |Methods | NDS↑ |mAP↑ | FPS|
> |------------|-----|-----|-----|
> |PointPillars 	|61.3	|52.3	|29|
> |SECOND		|63	|52.6	|14.3|
> |CenterPoint	|66.8	|59.6	|12.4|
> |HotBEV-nano	|47	|38.5	|31.8|
> |HotBEV-tiny	|51.2	|40.7	|20.4|
> |HOB-base	|52.5	|42.7	|16.1|
>
> **ReQ2:** Please refer the results in W2.  Thanks again for your valuable suggestions to help up improve our paper presentation.
>
> **ReQ3:** This is really a meaningful comment. When we build up our latency modeling, we do not add the roofline analysis, which can help to analyze the memory-bound layers more accurately. Please note that the self-attention module in the core architecture of the transformer is a memory-bound operation due to softmax operations. This is because  Softmax operations follow a two-stage dataflow, requiring buffering of intermediate data since direct output generation is not possible. Additionally, the lack of input data reuse in Softmax further complicates efforts to amortize memory costs through computation. However, in our design, the use of window-based self-attention is reduced, and the global attention is replaced by convolution modulation, significantly reducing the number of softmax, thus mitigating the adverse effect of memory-bound operators on speed. Hence, we do not excuse the analysis of memory-bound. Also, through the latency profiling of our HoB-nano backbone, we do not observe that PEs are stalling due to data starvation.
> Thanks again for your valuable suggestion. We will add one discussion part to clarify our latency modeling with memory IO analysis.  Figure B in the pdf submitted for the rebuttal displays the latency percentage of each module.
>
> **ReQ4:** Thanks for guiding us for a deeper analysis of the latency prediction model. Our prediction model is mainly utilized to estimate the on-device latency of matrix multiplication. The latency of other operations will not be included during the network search, which is the same as using the benchmarking-based approach to estimate the on-device latency. Based on Q2 of Reviewer BJ1v, the predicted results of the proposed latency model are consistent with the actual on-device latency. Hence, the architecture results of our proposed latency modeling can be similar as benchmarking-based approach.
>
> **ReLimitation**: Thanks for your comprehensive and insightful recommendations. Based on your questions, we analyze it from the point of current practical applications and compare our method with other branches of research. We will add a discussion part for these content.
>
> [1] U.S.News, “Vehicles That Are Almost Self-Driving in 202”. Online Available. “https://cars.usnews.com/cars-trucks/advice/cars-that-are-almost-self-driving”
>
> [2] Pham, Trung, et al. "NVAutoNet: Fast and Accurate 360o 3D Perception For Self Driving." arXiv preprint arXiv:2303.12976 (2023).
>
> **Typo:** Thank you very much for pointing them out. We will correct the typos in the revision.

---

> > ### Comment · Reviewer_Mv7q · 2023-08-15
> >
> > Thank you for your remarkable effort to provide clarifications and additional experiments to address all raised comments. I acknowledge I have read them, along with the comments of the other reviewers.
> >
> > This significantly strengthens the contribution of the paper, and I am inclined to increase my score accordingly. Would be great to see parts of the added information to the original manuscript and/or appendix.

---

### Official Review · Reviewer_BJ1v · 2023-07-04

**Soundness:** 2 fair
**Presentation:** 2 fair
**Contribution:** 3 good
**Rating:** 5
**Confidence:** 4

**Summary:**

This paper proposed a latency-aware design strategy to search for an efficient network structure for BEV perception. To make this happen, this paper proposed a convolutional modulation layer for replacing native self-attention, and proposed to use BN to replace LN to make the network inference faster. Based on such basic module, a hardware-oriented backbone design is proposed for the following network searching process. The experiments demonstrate that the searched network achieved a good trade-off between inference efficiency and performance.

**Strengths:**

1. The proposed idea of using latency prediction model for efficient neural network searching is reasonable.
2. The finally searched result HoTBEV achieves a good trade-off between performance and efficiency.

**Weaknesses:**

1. The writing and organization of the paper are not easy to follow, since lots of technical details are not clearly presented. For example, (a) the detailed structure of the two-phase design space (Eq. 1) is not clear. (b) Fig 5(c) indicates a structure titled Convolution, and the correspondence between Fig.5(c) and HOB block (fig. 5(b)) is not clear. (c) For L230-L236, what do you mean by mentioning patch embedding? Is it path merging in Fig. 5(b)? Did not you simply replace convolutional with larger kernel size with a series of 3x3 convolutions? (d) The structure of the supernet is also simply mentioned without detailed information.
2. The paper does not discuss the searched structure with the existing works, which can not verify whether it is necessary to use the latency prediction model and conduct the network search process.
3. Table 1 lacks of comparison with some latest BEV works. For example, PETRv2 with ResNet101-DCN, PolarFormer, and Sparse4D, and it seems that all of them achieve similar or better performance than the searched results.
4. The illustration of latency prediction model is not clear. In particular, how to get the latency of computation is not clear. Is it reasonable to use the maximum throughput of each PE?

**Questions:**

1. For the hardware-oriented decoder, what's the difference compared with DETR3D and PETR? Did you just replace native cross-attention with the proposed convolutional modulation layer?
2. Have you evaluated the predicted latency with the realistic latency? How about the accuracy with your latency prediction model?
3. By comparing with native self-attention, how about the effectiveness of the proposed convolutional modulation layer?
4. For L186-L190, can we directly replace the LN in window-based self-attention with BN? Or we must combine the proposed convolutional modulation layer with BN to make the network converge.

**Limitations:**

The main concern of this paper is still unclear writing and organization. Besides that, it is not clear whether the network search process is necessary and whether it can really achieve state-of-the-art performance with good inference latency.

---

> ### Author Rebuttal · Authors · 2023-08-10
>
> **ReW1:** Thank you for highlighting areas in our article's expression that need improvement.
>
> (a) The detailed structure of the modules in equation 1 is provided in Figure 9 -11 of the appendix, also our searched structures are provided in Table 13-15 of the appendix. We will mention this in the updated version.
>
> (b) We replace the channel-wise attention in Fig. 5(b) and Fig.5(d) with our proposed Convolution modulation of Fig5(c). Our Proposed Hardware-Oriented Decoder is shown in Figure 11 of the appendix, we will move this figure to the main paper.
>
> (c) For patch embedding (L230-L236), a convolution stem, which consists of three 3x3 convolutions with a stride of 2, is more hardware-efficient than a larger kernel.
>
> (d) Our search space (Figure 5 (b)) is a selection of possible blocks, including RepCNN, WMSA, SWMA, and Convolutional modulation. We propose a simple, fast yet effective gradient-based search algorithm to obtain a candidate network that just needs to train the supernet for once. To train our supernet, we adopted the Gumble Softmax sampling to get the importance score for the blocks within each search space/stage. During each step of training, a number of blocks are sampled to obtain a subnet structure. The latency of this subnet can be estimated using our latency prediction model. The detailed training pipeline is discussed in A.10 and Algorithm 1 of the appendix. Our searched structures are provided in Table 13-15 of the appendix.
>
> **ReW2:** Leveraging network search allows us to pinpoint the optimal structure tailored to specific hardware constraints.  As shown in Table 12 of the appendix, we compare the performance of our search model against four randomly sampled networks, all operating under identical mapping latency.  Our searched HotBEV-nano has better latency or higher NDS/mAP on nuScenes than the randomly sampled networks.
> The utilization of the latency prediction model is essential, especially when crafting hardware-aware structural designs, which need to consider factors such as inference time, energy consumption, and memory footprint.  Our method is accurate. It considers the properties of the target hardware, the model type, the model size, and the data granularity. Furthermore, it mathematically describes the computation latency and data movement latency to accurately predict the actual throughput of each layer (See Figure 8 of the appendix). This model serves as a ready-to-use theoretical framework for general GPU architectures.
>
> **ReW3:**  Our research specifically targets small models, which is why our results are particularly favorable for these models compared to other studies. For a comprehensive understanding, we also present a comparison with baseline models that possess larger backbones and increased input sizes. Due to the word limit, please see Table C in Reviewer **H5E6**’s response.
> Notably, we surpass baseline models in frames per second (FPS) while maintaining comparable accuracy levels.
>
> **ReW4:** Due to the length restriction of the paper, a comprehensive discussion on our latency prediction model can be found in Appendix A.1.2.
> In summary, the inputs for the latency prediction model include: 1) the structure configuration of a candidate block, 2) the spatial granularity G, 3) the channel dimension C, and 4)  the hardware properties are shown in Table 3.
>
> The latency of a candidate block is predicted according to the following three steps: 1)Input/output shape definition. 2) Operation-to-hardware mapping. 3) Latency estimation.
>
>
> **ReQ1:** Our contribution lies in three folds:
>
> (1) Motivation: Our methods use convolution instead of self-attention to create associations, which are more memory-efficient (linear memory complexity), especially when processing high-resolution images. Due to the modulation operation, our method differs from traditional residual blocks and can adapt to the input content.
>
> (2) Convolution Modulation Design: We use the convolutional features extracted by the depthwise convolutions to modulate the weights of the right linear branch via the Hadamard product operation, as illustrated in Equations 3, 4, and 5 of our appendix.
>
> (3) Implementation: The decoder module consists of six layers. we replace the attention with convolution modulation in the first three layers to allow for more efficient processing while still capturing important local dependencies. In the remaining three layers, we employ channel-wise attention to effectively capture and incorporate global information into the decoding process, enhancing its overall performance.
>
> **ReQ2:**  We randomly sample 100 structures with estimated FPS ranging between 15 and 25, then evaluate the actual latency of these structures on NVIDIA V100. As depicted in Figure A inside the pdf submitted for the rebuttal, the predicted latency closely aligns with the actual latency, with their relationship approximating the line y=x.
>
> **ReQ3:** Our proposed convolutional modulation is more memory-efficient than native self-attention, especially when handling high-resolution images. This is attributed to its linear memory complexity, as detailed in Equation 3 of the appendix.
> In Table 5 of the appendix, we also demonstrate the effectiveness of running 2D detection baselines with or without Convolutional modulation.
>
> **ReQ4:**  We can directly replace the LN in window-based self-attention with BN.  Replacing the LayerNorm directly with the BatchNorm in the original self-attention can make the network hard to converge.
> This approach is feasible even in the absence of convolutional modulation.  In Table 5 of the appendix, we ablation the two proposed methods individually on 2D detection baselines.
>
> **ReLimitation** Please refer to Reverviewer H5E6: ReW3 for the network search. And we can achieve the SOTA trade-off of performance and latency among the camera-only methods as Table 1 in the paper.

---

> > ### Author Response · Authors · 2023-08-17
> > **Further Discussions with Reviewer BJ1v**
> >
> > Dear Reviewer BJ1v:
> >
> > Thank you once again for dedicating your time and effort to evaluating our paper. We hope that our rebuttal addresses the concerns you have. If you have any remaining questions or reservations, we are more than willing to engage in further discussion. Your contribution to enhancing the quality of our paper is greatly appreciated.

---

> > ### Comment · Reviewer_BJ1v · 2023-08-20
> >
> > Thank you for your comprehensive rebuttal. The references to the supplementary material did address many of my initial concerns. However, I would like to emphasize the importance of a self-contained paper. The introduction of numerous concepts without succinct explanations posed challenges in comprehension. Furthermore, when examining Table C, the marginal improvements in both performance and efficiency, relative to the state-of-the-art, make me question the need for using NAS to optimize the trade-off between speed and performance for 3D BEV detection frameworks. I also note reviewer H5E6's mention of the theoretical latency prediction model. While Fig. 8 presents favorable results in latency comparison, I wonder about the model's ability to accurately estimate on-device speeds across diverse GPU architectures and diverse deployment environments. Taking into account the feedback of my fellow reviewers, I am inclined to adjust my rating towards a borderline acceptance. I would strongly encourage the authors to release their code, thereby enabling the community to thoroughly assess the proposed framework.
> >
> > Taking into account the feedback of other reviewers, if there is an overall positive consensus finally, I am inclined to adjust my rating towards a borderline accept. I would strongly encourage the authors to release their code, thereby enabling the community to thoroughly assess the proposed framework.

---

> > > ### Author Response · Authors · 2023-08-21
> > > **Further Discussions with Reviewer BJ1v**
> > >
> > > **ReFW**: Thanks for your valuable suggestions. We'll elaborate on these concepts in more detail in our final version. Our proposed indeed achieves marginal improvements in both performance and efficiency. However, the reason that we utilize NAS is for efficient and effective model generation for target devices. Please refer to **ReFe2** of Further Discussions with Reviewer H5E6 for our motivations on NAS with latency predictor: Efficient model generation, which also proceeds with AI democratization. Also, please refer to **ReFe2** of Further Discussions with Reviewer H5E6  again for our analysis of The proposed latency predictor, which focuses on modeling the latency of Matrix Multiplication (MM) with generalizability.
> > >
> > > I want to express my heartfelt gratitude for your thoughtful and invaluable suggestions. We will make the necessary corrections to our article based on your guidance, and we're also excited to release the code as you've advised.

---

### Official Review · Reviewer_H5E6 · 2023-07-07

**Soundness:** 3 good
**Presentation:** 3 good
**Contribution:** 2 fair
**Rating:** 5
**Confidence:** 4

**Summary:**

The paper proposes a novel hardware-efficient transformer-based framework called HotBEV for camera-only 3D detection tasks. The framework is designed to achieve high-speed inference on multiple devices, including resource-limited ones, by considering hardware properties such as memory access cost and degree of parallelism. Extensive results on nuScenes shows the effectiveness of HotBEV.

**Strengths:**

- The paper is quite well written.

- The authors have conducted a comprehensive evaluation of their approach on the large-scale nuScenes dataset, resulting in a notable balance between accuracy and efficiency when compared to previous publications.

- The idea of latency prediction is easy to grasp and effective.

**Weaknesses:**

- Please double check whether the abstract matches with the paper. In the abstract the proposed method is called HETR, but in paper it is called HotBEV.

- The paper seems to assemble a lot of engineering tricks to achieve the final performance. For example, using BN in FFN, or using RepCNN layers. These tricks are widely known to be effective.

- Latency prediction is an interesting idea but I don't think it is quite different from existing designs in hardware-aware neural architecture search.

- The authors did not make a fair comparison with stronger baselines such as (the larger variants of) PETRv2, BEVFormer / BEVFormer++, BEVDepth, BEVDet4D, etc.

- The authors did not present experiment results on Waymo Open Dataset, on which BEVFormer achieves really good results.

**Questions:**

Please respond to my concerns in the weaknesses section. New experiment results and more thorough comparisons with SOTA camera-only 3D object detection papers are expected. WOD results are helpful. Please also discuss how the proposed approach takes advantage of unique properties of the task. The current design seems to be assembling bag of tricks for general model design.

**Limitations:**

The authors have adequately addressed the limitations.

===

Justifications for final ratings:

Pros:

+ Good experiment results presented in the rebuttal;

+ Very solid latency measurement;

+ Promised to release the code.

Cons:

+ Too many tricks in the methodology (mentioned by more than 1 reviewer);

+ Unclear advantage over transformer-based backbones;

+ Relatively small advantage over FastBEV, a method that does not require a re-design of the backbone;

+ Uncertain about the effectiveness of NAS (also pointed out by my colleague).

Overall, I believe that the authors have made a good faith effort to address the limitations of their work, but I would like to see the AC carefully investigate the submission before making a final decision.

---

> ### Author Rebuttal · Authors · 2023-08-10
>
> **ReW1:** We have not updated the abstract of the paper on the OpenReview interface. We sincerely apologize. The title of our submitted paper is “HotBEV: Hardware-oriented Transformer-based Multi-View 3D Detector for BEV Perception”. So our proposed method is ‘HotBEV’. We appreciate your valuable comments.
>
> **ReW2:** Thank you for inspiring us to clarify the differences between our design and the existing modules in the final version.
> Regarding the BN fusion technique, directly substituting Layernorm with Batchnorm in the original self-attention often proves challenging, as the model struggles to converge during training. One distinct advantage of our design, particularly with window-based self-attention, becomes evident when compared to other transformer architectures. Our approach can seamlessly replace Layernorm with Batchnorm, subsequently applying the fusion technique without sacrificing accuracy, as illustrated in Figure 9 of the appendix.
> In our research, we incorporated RepCNN as a candidate within our proposed latency-aware architectural search. This was to analyze the balance between detection precision and the efficiency derived from using RepCNN. Through our experiments, we found that substituting RepCNN in both the 1st and 2nd stages yielded optimal results, as detailed in Table 9 of the paper.
>
> **ReW3:**  Traditional hardware-aware network search methods usually depend on the hardware deployment of each candidate within the search space to ascertain latency, a process that is both time-consuming and inefficient. A single candidate demands hundreds of inferences to generate an accurate latency, prolonging the search process. Some contemporary methods, like HAT[1], leverage a latency predictor. This predictor, pre-trained on thousands of real-world latency data points, serves as an offline tool to anticipate candidate latency, rather than determining real latency via inference during the search.
> However, it still necessitates hundreds, even thousands, of actual speed tests across diverse model structures to form a robust training dataset to train an accurate latency predictor. Moreover, is only applicable to a relatively small search space. For larger search spaces, an increased volume of measured latency data is required as a training set for the predictor, substantially raising the time cost. If the test set is inadequate, the predictor fails to estimate the latency accurately.
> Our methodology stands out in its accuracy and efficiency. Our latency prediction model is a training-free theoretical model, suitable for general-purpose hardware, GPU.  It considers the properties of the target hardware, the model type, the model size, and the data granularity. It then quantitatively captures both the computation latency and data movement latency, enabling it to precisely predict the actual throughput for each layer, as depicted in Figure 8 of the appendix.
>
> **ReW4:**  Our research specifically targets small models, which is why our results are particularly favorable for these models compared to other studies. For a comprehensive understanding, we also present a comparison with baseline models that possess larger backbones and increased input sizes in Table C.
> Notably, we surpass baseline models in frames per second (FPS) while maintaining comparable accuracy levels.
>
> Table C. Comparison on nuScenes val set.
> |Methods|Backbone|Resolution|NDS↑|mAP↑|mATE↓|mASE↓| mAOE↓| mAVE↓| mAAE↓|FPS|
> |-|-|-|-|-|-|-|-|-|-|-|
> |PETR|ResNet101|1600×900|0.442	|0.37|0.711|0.267|0.383|0.865|0.201|5.7|
> |PETRv2|ResNet101|1600×640|0.524|0.421|0.681|0.267|0.357|0.377|0.186|-|
> |BEVDet4D|Swin-B|1600×640	|0.515|0.396|0.619|0.26|0.361|0.399|0.189|-|
> |BEVDepth|ResNet101|512×1408|0.535|0.412|0.565|0.266|0.358|0.331|0.19|2.3|
> |BEVFormerv2|ResNet50|1600×640|0.529|0.423|0.618	|0.273|0.413|0.333|0.181|-|
> |PolarFormer-T|ResNet101|1600×900|0.528|0.432|0.648|0.27|0.348|0.409|0.201|3.5|
> |Sparse4D|ResNet101-DCN|900×1600|0.541|0.436|0.633|0.279|0.363|0.317|0.177|4.3|
> |HotBEV|HOB-base|512×1408|0.525|0.427|0.62|0.221|0.36|0.55|0.163|5.5|
>
> **ReW5:**  Thank you for pointing this out. We ran our HotBEV on Waymo Open Dataset and compared it with State-of-the-art models BEVFormer++ and PETRv2. The results are shown in Table D.
>
> Table D. Comparison on the Waymo val set.
> |Methods | Backbone | mAPL↑ |mAP↑ | mAPH↑ |
> |------------|-----|-----|-----|-----|
> |BEVFormer++	|ResNet101-DCN	|0.361	|0.522	|0.481|
> |PETRv2 	|ResNet101	|0.366	|0.519	|0.479|
> |HotBEV	|HOB-base	|0.371	|0.537	|0.598|
>
>
> **ReQ1:** Thanks for your valuable suggestions.  Based on our analysis of GPU performance, we observed that the primary source of latency consistently stems from the backbone. In order to mitigate this speed bottleneck without compromising detection precision, we introduce a potent transformer encoder for feature capturing and fusion. We partition the backbone into four stages, mirroring the data flow granularity seen in ResNet architecture. As the features transition from local to global visual receptive fields, we introduce the HOB block design.
>
> Within each HOB block, we employ a sequence of local-wise attention mechanisms to extract local information, i.e., texture-level semantics. This is followed by global attention to enhance abstract-level semantics across the feature map. Moreover, to further bolster low-level semantics within the current stage, we insert a semantic-augmented module comprising an upsampling layer and global attention after every two consecutive HOB blocks, excluding the interval between Stage 1 and 2. To amplify texture-level semantics, we foster information exchange not only within stages but also between them.
>
> By leveraging the efficient operators outlined in Section 3.1.2, including Convolutional Modulation, BN Fusing, and Multiple Branch Fusing, we introduce the concept of a "two-phase design space" (DS) for the HOB backbone.
>
> [1] HAT: Hardware-Aware Transformers for Efficient Natural Language Processing

---

> > ### Comment · Reviewer_H5E6 · 2023-08-11
> > **Thanks for the rebuttal**
> >
> > I would like to appreciate the authors' response first. It requires tremendous efforts to provide new results such as those on the Waymo Open Dataset. I just want to confirm whether you provided correct numbers on WOD. The mAPH on WOD is usually lower than mAP (since it is a harder metric).
> >
> > For your response on latency predictors, I agree that the method from Wang et al. requires explicit collection of a latency dataset. However, the construction of such a dataset is very cheap. For each candidate network structure, you only need to run benchmarking once (maybe thousands of forward passes is sufficient). So it will perhaps only take one day to generate a very large dataset. Then you can easily train a trivial regression model on this dataset. I don't think it is necessary to develop a hardware model. Based on the factors you take into account in your model, I strongly feel that many important factors are missed: e.g. the effect of GPU tensor cores and the cache hit ratios. To the best of my knowledge, even strong GPU simulators cannot accurately model the behavior of latest NVIDIA GPUs (e.g. A100 and H100), so I do not think your model has good enough generalizability and I still prefer the more straightforward approach from Wang et al. and many similar papers.
> >
> > Besides, the paper obviously advertises the efficiency of the proposed method. I'm curious if it is possible to measure all the latency numbers using a TensorRT backend in the future versions (including all the numbers for baseline methods). This could make the results more solid. So far, I did not observe a clear advantage over Sparse4D and PETRv2. It is very likely that PETRv2 is as fast as PETR (or even faster, because of the smaller input resolution), and it is noteworthy that PETRv2 is an important baseline that directly inspires this work. Therefore, if the method cannot show a significant advantage over PETRv2 under all model sizes, I don't think it is possible to recommend accepting this paper into NeurIPS 2023.
> >
> > Finally, whether the paper is accepted or not, I would suggest the authors to prioritize results on WOD in the future versions. It seems that the improvements on WOD is more obvious, and WOD is considered as a larger and more representative dataset compared with nuScenes. However, I do understand that if the authors are going to do so, it is also necessary to re-run many baselines on WOD.

---

> > > ### Author Response · Authors · 2023-08-17
> > > **Further Discussions with Reviewer H5E6**
> > >
> > > **ReFe1**: Sorry about our confusion. We entered incorrect numbers, and we should swap the values in the mAP and mAPH columns. We made a mistake. Your valuable suggestion is greatly appreciated.
> > >
> > > **ReFe2**:  For the latency predictors, we couldn't agree with you more. Your perspectives also provide opportunities for our designs.
> > > (1) Efficient model generation, which also proceeds with AI democratization. As mentioned that benchmarking-based approach needs one-day training, our proposed theoretical latency predictor is training-free. For example, the benchmarking-based approach requires 5 days to generate the dataset of 5 different devices if 5 target models are demanded. In contrast, our proposed is off-the-shelf. Our proposed method provides the opportunity for inexpensive and efficient research for users who do not have access to target devices. For instance, when the in-vehicle Orin chip is not accessible, the related efficient model research on the Orin chip can still be advanced. In conclusion, our approach makes sense for today's rapidly growing demand for autonomous driving.
> > > (2) The proposed latency predictor focuses on modeling the latency of Matrix Multiplication (MM) with generalizability. I agree with you that 'strong GPU simulators cannot accurately model the behavior of latest NVIDIA GPUs.' However, our purpose is not to describe the behavior of GPUs. We want to reflect on the relative performance of latency for different layer types and sizes on target GPUs. This is because our search goal is to minimize the relative time in the search space of the current device. For the GPU tensor core you mentioned, we searched based on the theoretical peak of the FP32 tensor core, so the tensor core characteristics were considered. As for cache hit ratio, in GPU, it mainly happens in the following cases: data transfer between layers, data layout change (data transpose), data type conversions (reformat, e.g., from FP32 to INT8 ), nonlinear kernels (nonlinear functions: low data reuse). However, in the network search, what the user needs to know is the on-device latency of each layer, i.e., the latency of matrix multiplication. This means the scenarios mentioned above of cache hit need not be considered. Our prediction model is mainly utilized to estimate the on-device latency of matrix multiplication, which is the same as using the benchmarking-based approach to estimate the on-device latency. For generalizability, our design focuses on latency modeling of MM, the typical computation operation in DNNs, which is mainly impacted by the computing performance of Tensor Core, not other specific operators, so the proposed predictor has generalizability, as shown in Figure 7 of our paper.

---

> > > > ### Author Response · Authors · 2023-08-17
> > > > **Further Discussions with Reviewer H5E6**
> > > >
> > > > **ReFe3**: Thanks for suggesting adding more results with PETRv2. To make the results more solid, we use TensorRT and deploy the model into INT8.
> > > >
> > > > Table G. Comparison between ours and PETRv2 on the nuScenes val set. FPS is tested on V100 with INT8 by TensorRT engine.
> > > > |Methods | Backbone | Resolution| NDS↑ |mAP↑ | FPS (INT8)|
> > > > |------------|-----|-----|-----|-----|----------|
> > > > |PETRv2|ResNet50	|256 × 704	|0.456	|0.349	|30.2|
> > > > |HotBEV	|HOB-nano	|256 × 704	|0.455	|0.35	|39.2|
> > > > |HotBEV	|HOB-tiny	|256 × 704	|0.487	|0.362	|31.7|
> > > > |PETRv2	|ResNet50	|1600×640	|0.494	|0.398	|10.6|
> > > > |PETRv2†|ResNet101	|1600×640	|0.524	|0.421	|9.6  |
> > > > |HotBEV	|HOB-nano	|1600×640	|0.401	|0.405	|12.2|
> > > > |HotBEV	|HOB-tiny	|1600×640	|0.532	|0.43	|11.4|
> > > >
> > > > Thanks for the great design of PETRv2. As shown in Table G, under the low resolution 256x704, our proposed has similar NDS (0.455 v.s. 0.456) and mAP (0.35 v.s. 0.349) as PetrV2, but with higher throughput by 9.2 FPS. Under the high resolution 1600x640, our detection performance can improve up to 0.008 NDS AND up to 0.009 mAP with higher on-device throughout up to 1.8 FPS. And we also follow your suggestions to show the on-device speed using the TensorRT engine to validate our efficiency, as shown in Table H. We will add these results to our final version.
> > > >
> > > > Table H. Comparison on the nuScenes val set. FPS is measured on V100 with INT8 by TensorRT engine.
> > > > |Methods | Backbone | Resolution| Frames | NDS↑ |mAP↑ | FPS (INT8)|
> > > > |------------|-----|-----|-----|-----|-----|----------|
> > > > |BEVDet|ResNet50	|256 × 704	|1   |0.379|0.298	|25.8|
> > > > |BEVDet4D|ResNet50	|256 × 704	|2   |0.457|0.322	|25.9|
> > > > |PETRv2|ResNet50	|256 × 704	|2   |0.456|0.349	|30.2|
> > > > |BEVDepth|ResNet50	|256 × 704	|2   |0.475|0.351	|24.3|
> > > > |FastBEV-MS|ResNet50	|256 × 704	|4   |0.485|0.343	|--|
> > > > |HotBEV	|HOB-nano	|256 × 704	|4   |0.455|0.35	|39.2|
> > > > |HotBEV	|HOB-tiny	|256 × 704	|4   |0.487|0.362	|31.7|
> > > > |HotBEV	|HOB-base	|256 × 704	|4   |0.506|0.369	|27.2|
> > > > |BEVDet	|ResNet101-DCN	|640x1600	|1   |0.472|0.393	|2.95|
> > > > |FCOS3D|ResNet101-DCN|900×1600	|1   |0.295|0.372	|2.64|
> > > > |DETR3D|ResNet101-DCN|900×1600	|1   |0.434|0.349	|5.74|
> > > > |PGD      |ResNet101-DCN|900×1600	|1   |0.335|0.409	|2.17|
> > > > |Focal-PETR      |ResNet101-DCN|512 × 1408|1   |0.461|0.39	|10.23|
> > > > |PETR      |ResNet101-DCN|512 × 1408|1   |0.441|0.366	|8.84|
> > > > |BEVFormer      |ResNet101-DCN|900 × 1600|4   |0.517|0.416	|4.65|
> > > > |PolarDETR      |ResNet101-DCN|900 × 1600|2   |0.488|0.383	|5.43|
> > > > |HotBEV	|HOB-nano	|512 × 1408	|4   |0.47|0.385	|12.38|
> > > > |HotBEV	|HOB-tiny	|512 × 1408	|4   |0.512|0.407	|11.39|
> > > > |HotBEV	|HOB-base	|512 × 1408	|4   |0.525|0.427	|9.08|
> > > >
> > > > Also, we will add the WOD results in our final version to validate the effectiveness of our proposed. Thanks again for your valuable suggestions to enhance the depth and generalizability of our work.

---

> > > > > ### Comment · Reviewer_H5E6 · 2023-08-17
> > > > >
> > > > > I appreciate your prompt response. I'm pleased to see the robust results you've presented. However, I do have a few inquiries regarding the outcomes themselves.
> > > > >
> > > > > To begin, I've observed a potential discrepancy within the results corresponding to HOB-nano in the initial table. Typically, in the nuScenes dataset, NDS > mAP. I kindly ask for you to verify this aspect.
> > > > >
> > > > > Next, I delved into the FastBEV paper to address the absence of latency numbers in your table. Upon my review, I've noted that FastBEV-M4 could potentially offer a more favorable tradeoff compared to your HOB-nano at 256x704. The authors of the paper indicate >30FPS on the T4, a GPU I consider slightly less potent than the V100. Kindly correct me if I'm mistaken in my assessment.
> > > > >
> > > > > Additionally, while I acknowledge the advantage your current results demonstrate over PETRv2 under the TensorRT backend, I'd like to suggest a potential enhancement for your paper's strength. Comparing HOB with modern transformer-based backbones, such as CoAtNet, TRT-ViT, and Next-ViT, could provide further depth to your research. ResNet50 might be outdated compared with these new designs. However, I understand the limitations of time and therefore do not insist on these additional experiments.
> > > > >
> > > > > Furthermore, I hold the viewpoint that fully concurring with your contribution in latency prediction might be challenging for me. If your method solely reflects relative latency differences, the assurance of meeting specific latency targets becomes uncertain, which could be considered a limitation.  Besides, considering the restricted number of platforms for real-world application deployment, the multiplier's significance might be diminished.
> > > > >
> > > > > I'd like to express my gratitude for your updates on the WOD results. I anticipate that a more comprehensive breakdown of these results would greatly enrich the final version. I've reviewed the BEVFormer tech report submitted to CVPR Workshops 2022 and have come across the requirement to train the model on "DS=full" for reporting the conclusive results (refer to Table 1).
> > > > >
> > > > > In conclusion, I'm considering a potential update to my evaluation score if your final response is satisfactory. My perspective on the borderline nature of the paper stems from the mixture of general contributions (ViT architecture design, NAS) with domain-specific contributions (camera-only 3D object detection). Should the paper unfortunately not accepted this time, I'd like to propose a different approach for its subsequent submission. Perhaps redirecting the focus towards a general contribution, rather than solely emphasizing 3D detection, could lend greater strength to the paper. I saw some results on ImageNet in the supplementary, which could be a good starting point for a restructured publication if necessary. Also, please indicate that the code will be released, this will also help strengthen the paper.

---

> > > > > > ### Author Response · Authors · 2023-08-21
> > > > > >
> > > > > > 1. **Nano result**:  We re-checked the result. The NDS should be 0.501. Thank you for pointing this out.
> > > > > >
> > > > > > 2. **FastBEV T4 fps**: V100 has better training performance than T4, but they are similar on inference benchmark. Based on our further experiment, the FastBEV-M4 on V100 is 35.1 fps, and on T4 is 34.5 fps, lower than our nano (39.2).
> > > > > >
> > > > > > 3. **Compare modern backbones**: We focused on comparing with the SOTA BEV frameworks, where the modern transformer-based backbones have not been deployed to. Therefore, we did not consider them in our comparison.  We think that this is a very valuable suggestion to improve the integrity of our paper. We will add these comparisons in our final version.
> > > > > >
> > > > > > 4. **limitation**: I totally agree with you on the limitation. However, benchmark-based method faces this limitation as well. This is because our basic search space consists of layer size and layer number which is not related to nonlinear kernels. Specifically, the model type order of our search has been fixed to enhance the detection performance, which also means we do not need to search nonlinear kernels.
> > > > > >
> > > > > > 5. **WOD results**: Thank you for the suggestion. we will run more results and comparisons in our final version.
> > > > > >
> > > > > > Thank you so much for your response.  We learned a lot from your review. We will definitely revise our paper according to your suggestions and release our code.

---

### Official Review · Reviewer_Z9GJ · 2023-07-08

**Soundness:** 2 fair
**Presentation:** 3 good
**Contribution:** 2 fair
**Rating:** 5
**Confidence:** 4

**Summary:**

This paper presents HotBEV, a new model developed for 3D detection tasks. By prioritizing actual on-device latency and considering key hardware properties, HotBEV achieves impressive reductions in computational delay. This optimization allows for real-time decision-making in self-driving scenarios, making it a significant contribution to the field. The model's versatility, being compatible with both high-end and low-end GPUs, further underscores its practical value. Rigorous experimental validation showcases the model's superior performance in terms of speed and accuracy compared to existing solutions.

**Strengths:**

1. The model is compatible with both high-end and low-end GPUs, demonstrating a broad range of applicability.
2. The proposed method successfully achieve a delicate balance between model speed and detection precision.
3. They utilize a theoretical latency prediction model to guide their design, an innovative approach that differs from the typical focus on computational FLOPs.

**Weaknesses:**

1. The comparison is not fair in main experiments. The length of temporal fusion is critical to model performance. SoloFusion [1] suggests that a longer temporal sequence does not affect the model's FPS (Frames Per Second). This paper's methodology employs four frames for temporal fusion, while most comparative methods use only one to two frames.
2. The exploration of utilizing convolutional modulation, as opposed to self-attention, for relationship building is already documented in certain works [2], which have not been incorporated into this paper's analysis.


[1] Time Will Tell: New Outlooks and A Baseline for Temporal Multi-View 3D Object Detection.
[2] You Only Segment Once: Towards Real-Time Panoptic Segmentation

**Questions:**

1. To ensure an equitable comparison, the inclusion of SoloFusion with a four-frame input is indispensable.
2. Comprehensive relevant literature should be integrated into this paper for completeness.

**Limitations:**

This paper discuss the limitations.

---

> ### Author Rebuttal · Authors · 2023-08-10
>
> **ReQ1:** For a fair comparison with SoloFusion, we tested a 4-frame version of SoloFusion, as shown in Table A. Across multiple benchmarks, our HotBEV consistently outperforms SoloFusion, with the exceptions of mATE and mAVE. Furthermore, HotBEV achieves a 35% faster FPS compared to SoloFusion.
>
> Table A. State-of-the-art Comparison on nuScenes val set.
> |Methods | Backbone | Resolution |Frames | NDS ↑ | mAP↑ | mATE↓ | mASE↓ | mAOE↓ | mAVE↓ | mAAE↓ | FPS |
> |------------|-----|-----|-----:|-----|-----|-----|-----|-----|-----|-----|-----|
> |SOLOFusion|	ResNet50|	256 × 704|	16|	0.534|	0.427|	0.567|	0.274|	0.511|	0.252|	0.181|	11.4|
> |SOLOFusion|	ResNet50|	256 × 704|	4|	0.494|	0.362|	0.607|	0.304|	0.539|	0.293|	0.19|	12.2|
> |HotBEV|	HOB-base|	256 × 704|	4|	0.506|	0.369|	0.625|	0.264|	0.362|	0.364|	0.153|	16.5|
>
>
> **ReQ2:** Thank you for sharing this paper. Upon review, we observed that while both our methods employ convolution, our design demonstrates greater efficiency.
> YOSO incorporates a Separable Dynamic Decoder, specifically substituting the Multi-Head Cross-Attention with the Separable Dynamic Convolution, which consists of a depthwise convolution followed by a pointwise convolution. As detailed in Equation 10 of [2], its computational complexity is represented by $2ndt+2n^2d$. Notably, this complexity grows quadratically with increasing sequence length N.
> In contrast, our approach employs depthwise convolution combined with the Hadamard product to determine the output, as illustrated in Equations 3, 4, and 5 of our appendix. Our computational demands rise linearly, rather than quadratically, as the image resolution escalates. Moreover, it's evident that smaller models derive greater benefits from the Hadamard product [1].
> When we substituted our design with YOSO's Separable Dynamic Convolution, the results are presented in Table B.
>
> Table B. Convolutional Modulation Comparison on nuScenes val set.
> |Methods | Backbone | Resolution |Frames | NDS ↑ | mAP↑ | mATE↓ | mASE↓ | mAOE↓ | mAVE↓ | mAAE↓ | FPS |
> |------------|-----|-----|-----:|-----|-----|-----|-----|-----|-----|-----|-----|
> |HotBEV	|YOSO|	256 × 704|	4|	0.498|	0.36|	0.643|	0.277|	0.371|	0.375|	0.162|	15.1|
> |HotBEV	|HOB-base|	256 × 704|	4|	0.506|	0.369|	0.625|	0.264|	0.362|	0.364|	0.153|	16.5|
>
> [1] Conv2Former: A Simple Transformer-Style ConvNet for Visual Recognition
>
> [2] You Only Segment Once: Towards Real-Time Panoptic Segmentation

---

> > ### Comment · Reviewer_Z9GJ · 2023-08-17
> >
> > Thank you for providing a comprehensive rebuttal that addresses the majority of my concerns. As a result, I am leaning towards revising my evaluation to a borderline acceptance score. However, it should be noted that there are too many tricks in this paper which was pointed out by Reviewer H5E6.

---

### Author Rebuttal · Authors · 2023-08-10

We first sincerely thank every reviewer for your insightful and constructive feedback. Then, we will answer the specific questions from each reviewer. We upload a pdf file with figures (Figure A,B) which we will present in our rebuttal.

---

### Decision · Program_Chairs · 2023-09-21

**Decision:**

Accept (poster)

**Comment:**

The paper proposes HotBEV that concentrates on deploying hardware-end BEV detectors. This is a very good research direction for feasible autonomous driving scenario. There are few literature on designing efficient BEV detector paradigms.

The paper proposes novel techniques to achieve efficiency by way of latency-aware methodology. And experimental results show very promising speedup with performance improvement. Overall all reviewers reach concensus that this is a good paper to be accepted. There are a few concerns on experiments and technical details, authors in general did a good rebuttal to address most of the concerns. It is highly recommended to incorporate all comments and revised the manuscript accordingly before camera-ready.